# Boosting Methods for Interval-censored Data with Regression and Classification

**Yuan Bian**[1], **Grace Y. Yi**[1,2,3,*] **Wenqing He**[1]
[1]Department of Statistical and Actuarial Sciences, University of Western Ontario
[2]Department of Computer Science, University of Western Ontario
[3]Vector Institute, Canada
yb2620@cumc.columbia.edu, {gyi5, whe23}@uwo.ca

## Abstract

Boosting has garnered significant interest across both machine learning and statistical communities. Traditional boosting algorithms, designed for fully observed random samples, often struggle with real-world problems, particularly with interval-censored data. This type of data is common in survival analysis and time-to-event studies where exact event times are unobserved but fall within known intervals. Effective handling of such data is crucial in fields like medical research, reliability engineering, and social sciences. In this work, we introduce novel nonparametric boosting methods for regression and classification tasks with interval-censored data. Our approaches leverage censoring unbiased transformations to adjust loss functions and impute transformed responses while maintaining model accuracy. Implemented via functional gradient descent, these methods ensure scalability and adaptability. We rigorously establish their theoretical properties, including optimality and mean squared error trade-offs. Our proposed methods not only offer a robust framework for enhancing predictive accuracy in domains where interval-censored data are common but also complement existing work, expanding the applicability of existing boosting techniques. Empirical studies demonstrate robust performance across various finite-sample scenarios, highlighting the practical utility of our approaches.

## 1 Introduction

Boosting (Schapire, 1990; Freund, 1995) is a foundational technique in machine learning, transforming weak learners into strong learners through iterative refinement (Schapire & Freund, 2012). This iterative nature not only increases predictive accuracy (Quinlan, 1996; Bauer & Kohavi, 1999; Dietterich, 2000) but also enhances robustness against overfitting (Bühlmann & Hothorn, 2007; Schapire & Freund, 2012), making boosting a popular choice for various applications. The *AdaBoost* algorithm (Freund & Schapire, 1996) was a groundbreaking development and remains a highly effective off-the-shelf classifier (Breiman, 1998). Subsequent research (Breiman, 1998; 1999; Mason et al., 1999) revealed that AdaBoost can be viewed as a steepest descent algorithm in a function space defined by base learners. Boosting continued to grow as Friedman et al. (2000) and Friedman (2001) extended its application to regression and multiclass classification within a broader statistical framework, and it is interpreted as a method of function estimation. In this expanded context, Bühlmann & Yu (2003) introduced $L_2Boost$, a computationally efficient boosting algorithm that leverages the $L_2$ loss function. More recently, Chen & Guestrin (2016) proposed *XGBoost*, a scalable and useful tree boosting system, and Ke et al. (2017) introduced *LightGBM*, an efficient tree boosting algorithm.

Despite the success of boosting methods, a key limitation persists: traditional boosting algorithms assume access to a fully observed random sample of data. In many real-world applications, however, data are incomplete or censored. This issue is particularly pronounced in fields like survival analysis, where interval-censored data are becoming increasingly prevalent.

---

*Corresponding author

## 1.1 LITERATURE REVIEW

Recent research in boosting has focused on handling incomplete or censored data. Most efforts have extended boosting methods to accommodate right-censored responses (e.g., Ridgeway, 1999; Hothorn et al., 2006; Wang & Wang, 2010; Mayr & Schmid, 2014; Bellot & van der Schaar, 2018; Yue et al., 2018; Bellot & van der Schaar, 2019; Barnwal et al., 2022; Chen & Yi, 2024) or missing responses (e.g., Bian et al., 2025a;b). In theses cases, techniques like imputation and weighting are employed to construct unbiased loss functions for training.

While these approaches have addressed some issues related to incomplete data, a significant gap remains in handling interval-censored data, where event times are known only to lie within specific intervals. This scenario, prevalent in survival analysis (e.g., Sun, 2006), is more complex than right censoring, as the response variable is completely unobserved though known to fall within an interval. Research on interval-censored data has expanded across various domains. For example, Yao et al. (2021) introduced a survival forest method using the conditional inference framework, while Cho et al. (2022) developed the *interval censored recursive forests* method for non-parametric estimation of the survivor functions. Yang et al. (2024) leveraged the *censoring unbiased transformation* (Fan & Gijbels, 1994; 1996) to create tree algorithms designed for interval-censored data. However, these approaches do not capitalize on the strengths of boosting, which could significantly enhance predictive performance and robustness.

## 1.2 OUR CONTRIBUTIONS

We propose a framework that extends boosting methods to address interval-censored data, a critical yet underexplored problem in machine learning. Our contributions enhance the applicability of boosting algorithms to complex censoring structures:

- We propose two methods, called L2Boost-CUT and L2Boost-IMP, to extend boosting for handling interval-censored data. L2Boost-CUT adjusts the loss function with the censoring unbiased transformation (CUT), while L2Boost-IMP uses an imputation-based approach leveraging CUT. Both methods handle interval-censoring flexibly, avoiding restrictive assumptions and enabling predictions of survival time, probability, and status.

- We provide a rigorous theoretical analysis of our methods, evaluating their mean squared error (MSE), variance, and bias, as well as the connection between the two proposed methods. Our results demonstrate that by incorporating smoothing splines as base learners, the proposed framework achieves optimal MSE rates in both regression and classification tasks. These insights extend the understanding of boosting methods, building upon and generalizing the foundational results from Bühlmann & Yu (2003) for complete data.

- We validate our methods through extensive experiments on both synthetic and real-world datasets. Results show that $L_2$Boost-CUT and $L_2$Boost-IMP offer robust and scalable solutions for handling interval-censored data and enhancing the generalizability of boosting algorithms.

## 2 PRELIMINARIES

Let $Y$ denote the survival time of an individual, and let $X$ denote the associated $p$-dimensional feature vector, where $Y \in \mathbb{R}^+$ and $X \in \mathcal{X}$, with $\mathbb{R}^+$ representing the set of all positive real values and $\mathcal{X}$ denoting the feature space. Our objective is to learn a predictive model that well predicts a transformed target variable $g(Y)$, where $g$ is a user-defined transformation and $g(Y) \in \mathcal{Y}$, with $\mathcal{Y} \subseteq \mathbb{R}$. The choice of $g$ depends on the task of interest. For instance, setting $g(Y) = Y$ directly models the survival time; setting $g(Y) = \log Y$ removes the positivity constraint of $Y$. For binary classification tasks, we can set $g(Y) = 2I(Y > s) - 1$ to predict the survival status at time $s$, where $s$ is a prespecified threshold and $I$ is the indicator function.

We define the hypothesis space, $\mathcal{F} = \{f : \mathcal{X} \to \mathcal{Y}\}$, consisting of real valued functions. Let $\mathcal{Y}^d$ denote $\mathcal{Y} \times \ldots \times \mathcal{Y} \triangleq \{(y_1, \ldots, y_d) : y_j \in \mathcal{Y} \text{ for } j = 1, \ldots, d\}$ for a positive integer $d$. Let $L : \mathcal{Y}^2 \to \mathbb{R}_{\geq 0}$ denote a loss function, which quantifies the error between the predicted and true

values, where $\mathbb{R}_{\geq 0} = \mathbb{R}^+ \cup \{0\}$. For $f \in \mathcal{F}$, define the *expected loss*, or *risk* as

$$R(f) = E\{L(Y, f(X))\}, \tag{1}$$

where the expectation is taken with respect to the joint distribution of $X$ and $Y$. The goal is to find the optimal function $f^*$ that minimizes the risk:

$$f^* = \arg\min_{f \in \mathcal{F}} R(f),$$

assuming its existence and uniqueness.

In practice, the joint distribution of $X$ and $Y$ is unknown, and we only have access to a finite sample of $n$ independent observations of $X$ and $Y$, say $\mathcal{O}_c \triangleq \{\{X_i, Y_i\} : i = 1, \dots, n\}$. For simplicity, we use uppercase letters $X$, $Y$, $X_i$, and $Y_i$, with $i = 1, \dots, n$, to represent both random variables and their realizations. To approximate $f^*$, we minimize the *empirical risk*, which serves as proxy for the expected loss:

$$\hat{f}_c = \arg\min_{f \in \mathcal{F}} \left\{ n^{-1} \sum_{i=1}^{n} L(Y_i, f(X_i)) \right\}. \tag{2}$$

In the absence of censoring, where survival times $Y_i$ are fully observed for all study subjects, $\hat{f}_c$ can be obtained using a boosting algorithm that iteratively improves base learners. Specifically, the $L_2$Boost algorithm, a variant of boosting using the $L_2$ loss function, minimizes the empirical risk via steepest gradient descent to iteratively refine the estimates of $\hat{f}_c$. At iteration $t$, given the current estimate $f^{(t-1)}$, the algorithm updates the model by adding an increment term, denoted $\hat{h}^{(t)}$, to form the updated estimate $f^{(t)}$:

$$f^{(t)} = f^{(t-1)} + \hat{h}^{(t)}, \tag{3}$$

where $\hat{h}^{(t)}$ is a function mapping from $\mathcal{X}$ to $\mathcal{Y}$, called a base learner, determined by

$$\hat{h}^{(t)} = \arg\min_{h^{(t)}} \left[ n^{-1} \sum_{i=1}^{n} \left\{ -\partial L\left(Y_i, f^{(t-1)}(X_i)\right) - h^{(t)}(X_i) \right\}^2 \right], \tag{4}$$

with $\partial L\left(Y_i, f^{(t-1)}(X_i)\right) \triangleq \left. \frac{\partial L(u,v)}{\partial v} \right|_{u=Y_i, v=f^{(t-1)}(X_i)}$ for $i = 1, \dots, n$. Here, $\hat{h}^{(t)}$ in (4) can be interpreted as the least squares estimate of $E\left(-\partial L\left(Y_i, f^{(t-1)}(X_i)\right) \big| X_i\right)$. Thus, the $L_2$Boost algorithm can be seen as repeated least squares fitting of residuals (Friedman, 2001). At a stopping iteration $\tilde{t}$, determined by a suitable stopping criterion, the final estimator of $f$ is given by

$$\hat{f}_c \triangleq f^{(\tilde{t})} = f^{(0)} + \sum_{j=1}^{\tilde{t}} \hat{h}^{(j)},$$

where $f^{(0)}$ represents an initial value for estimating $f$.

On the other hand, for classification tasks, particularly when the response $g(Y)$ is a step function, e.g., $g_s(Y) = 2I(Y > s) - 1$ for a given $s$, which maps the response $Y$ to the set $\{-1, 1\}$, the $L_2$Boost algorithm can be modified as "$L_2$ Boost with constraints" ($L_2$WCBoost) algorithm (Bühlmann & Yu, 2003). This modification allows us to handle binary classification problems, where the goal is to approximate $E\{g_s(Y_i)|X_i\}$, given by $2E\{I(Y_i > s)|X_i\} - 1$. In this case, $f^{(t)}$ in (3) is revised as:

$$f^{(t)} = \text{sign}\left(\tilde{f}^{(t)}\right) \min\left(1, \left|\tilde{f}^{(t)}\right|\right), \text{ with } \tilde{f}^{(t)} = f^{(t-1)} + \hat{h}^{(t)}, \tag{5}$$

where $\text{sign}(u) = -1$ if $u < 0$, 0 if $u = 0$, and 1 if $u > 0$. The modification of $\tilde{f}^{(t)}$ with the sign function included in (5) ensures that the final estimate $f^{(t)}$ stays within the range $[-1, 1]$, resulting in a meaningful output for binary classification.

## 3 PROBLEM AND METHODOLOGY

### 3.1 INTERVAL-CENSORED DATA

Interval censoring occurs when, instead of directly observing the exact survival time $Y$, we only observe a pair of time points $\{L, R\}$ such that $Y$ lies within the interval $(L, R]$, where $0 \leq L <$

$R \leq \infty$. Different scenarios arise depending on the values of $L$ and $R$: $L = 0$ yields a left-censored observation; $R = \infty$ leads to a right censored observation; $0 < L < R < \infty$ gives a truly interval-censored observation; and when $L = Y^-$ and $R = Y$, we have the exact observation, where $Y^- \triangleq \lim_{a \to 0^+}(Y - a)$, with $a \to 0^+$ representing $a$ approaching 0 from the positive side. Let $[0, \tau]$ denote the study period, with $\tau$ being finite. Following standard practice for modeling interval-censored data (e.g., Zhang et al., 2005; Cho et al., 2022), we assume *conditionally independent interval censoring*, meaning that given features $X$, as well as $L = l$, $R = r$, $L < Y \leq R$, the probability of the survival time $Y$ occurring before a positive value depends only on $l < Y \leq r$. Formally,

$$\Pr(Y < y | L = l, R = r, L < Y \leq R, X) = \Pr(Y < y | l < Y \leq r, X),$$

for any positive $y, l$, and $r$, with $l < r$.

Suppose for subject $i = 1, \ldots, n$, there are $M$ observation times $u_{i,1} < u_{i,2} < \ldots < u_{i,M} < \infty$ beyond $u_{i,0} \triangleq 0$, where $M$ is a random integer, with $m$ denoting its realization. While the randomness of $M$ does not affect calculations for a given dataset, its presence reflects real-world data uncertainty with varying numbers of observations. For a dataset with $m \geq 2$ and $i = 1, \ldots, n$, define the censoring indicators for each subject $i$ and interval $j$ as $\Delta_{i,j} \triangleq I(u_{i,j-1} < Y_i \leq u_{i,j})$ for $j = 1, \ldots, m$, and $\Delta_{i,m+1} \triangleq I(Y_i > u_{i,m})$. Clearly, $\Delta_{i,m+1} = 1 - \sum_{j=1}^{m} \Delta_{i,j}$. These indicators reflect whether the true survival time $Y_i$ falls within the corresponding time interval. Let the observed data for subject $i$ be $\mathcal{O}_i \triangleq \{\{X_i, u_{i,j}, \Delta_{i,j}\} : j = 1, \ldots, m\}$, and let the full observed dataset be $\mathcal{O}^{\text{IC}} \triangleq \cup_{i=1}^{n} \mathcal{O}_i$. Alternatively, $\mathcal{O}^{\text{IC}}$ can be equivalent expressed as $\mathcal{O}^{\text{IC}} = \{\{X_i, L_i, R_i\} : i = 1, \ldots, n\}$, where for each subject $i$, we identify the interval $(L_i, R_i]$ containing $Y_i$ by finding the index $j_i \in \{1, \ldots, m\}$ such that $u_{i,j_i-1} \leq Y_i \leq u_{i,j_i}$, with $L_i = u_{i,j_i-1}$, $R_i = u_{i,j_i}$. The sequence $\{\{L_i, R_i\} : i = 1, \ldots, n\}$ is then ordered in increasing order and the distinct values are denoted as $v_1 < v_2 < \ldots < v_{m_v}$.

## 3.2 BOOSTING LEARNING WITH INTERVAL-CENSORED DATA

To address the interval censoring effects, we modify the initial loss function $L(g(Y_i), f(X_i))$ to be an adjusted loss function $L^*(\mathcal{O}_i, f(X_i))$, with the argument $g(Y_i)$ in $L$ replaced by $\mathcal{O}_i$ in $L^*$, and further, we require:

$$E\{L^*(\mathcal{O}_i, f(X_i))\} = E\{L(g(Y_i), f(X_i))\}. \tag{6}$$

This means that minimizing the expected adjusted loss $E\{L^*(\mathcal{O}_i, f(X_i))\}$ is equivalent to minimizing the original risk function $R(f)$ defined in (1), treating $Y_i$ as if it were not interval censored but available. Here, we focus on the $L_2$ loss function, expressed as:

$$L(g(Y_i), f(X_i)) = \frac{1}{2}\{g(Y_i)\}^2 - g(Y_i)f(X_i) + \frac{1}{2}\{f(X_i)\}^2. \tag{7}$$

For $k = 1, 2$, we adjust $\{g(Y_i)\}^k$ using the transformation:

$$\tilde{Y}_k(\mathcal{O}_i) \triangleq \sum_{j=1}^{m} \Delta_{i,j} E\left(\{g(Y_i)\}^k \big| \Delta_{i,j} = 1, X_i\right), \tag{8}$$

where

$$E\left(\{g(Y_i)\}^k \big| \Delta_{i,j} = 1, X_i\right) = \frac{1}{S(u_{i,j}|X_i) - S(u_{i,j-1}|X_i)} \int_{u_{i,j-1}}^{u_{i,j}} \{g(y)\}^k dS(y|X_i) \tag{9}$$

for $j = 1, \ldots, m$, and $S(y|X_i)$ represents the conditional survivor function of $Y_i$ given $X_i$.

We propose a modified version for (7), called the *censoring unbiased transformation* (CUT)-based $L_2$ loss function, given by

$$L_{\text{CUT}}(\mathcal{O}_i, f(X_i)) = \frac{1}{2}\tilde{Y}_2(\mathcal{O}_i) - \tilde{Y}_1(\mathcal{O}_i)f(X_i) + \frac{1}{2}\{f(X_i)\}^2. \tag{10}$$

**Proposition 1.** *For the proposed CUT-based loss function (10), we have*

$$E\{L_{CUT}(\mathcal{O}_i, f(X_i))\} = E\{L(Y_i, f(X_i))\}.$$

This proposition ensures the validity of the CUT-based loss function (10), as it leads to the same risk (1) as that of the original loss function. Consequently, (2) can be implemented with the loss function replaced by (10), where for $k = 1, 2$, $\hat{Y}_k(\mathcal{O}_i)$ in (8) is replaced by its estimate, denoted $\hat{Y}_k(\mathcal{O}_i)$, which is derived from replacing $S(y|X_i)$ with its estimate (to be described in Section 3.3). Let $\hat{L}(\mathcal{O}_i, f(X_i))$ denote the resulting estimate of (10), and let $\hat{f}_n^{\mathrm{CUT}}$ denote a resulting estimate of (2) with $L(Y_i, f(X_i))$ replaced by $\hat{L}(\mathcal{O}_i, f(X_i))$.

Algorithm 1 outlines a pseudo-code for obtaining $\hat{f}_n^{\mathrm{CUT}}$; its implementation code is available on GitHub at `https://github.com/krisyuanbian/L2BOOST-IC`. The algorithm modifies the usual $L_2$Boost algorithm (Bühlmann & Yu, 2003) for (2), with the initial $L_2$ loss function $L(Y_i, f(X_i))$ replaced by $\hat{L}(\mathcal{O}_i, f(X_i))$, which directly applies to interval-censored data. Alternatively, one may employ the usual $L_2$Boost algorithm, but replace unobserved $Y_i$ with $\hat{Y}_1(\mathcal{O}_i)$. Specifically, (12) on Line 7 of Algorithm 1 is replaced by

$$\left| n^{-1} \sum_{i=1}^{n} L\left(\hat{Y}_1(\mathcal{O}_i), f^{(\tilde{t})}(X_i)\right) - n^{-1} \sum_{i=1}^{n} L\left(\hat{Y}_1(\mathcal{O}_i), f^{(\tilde{t}-1)}(X_i)\right) \right| \leq \eta,$$

together with replacing $\hat{L}(\mathcal{O}_i, \cdot)$ on Lines 3 and 4 of Algorithm 1 by $L\left(\hat{Y}_1(\mathcal{O}_i), \cdot\right)$. We refer to these two algorithms as $L_2$Boost-CUT and $L_2$Boost-IMP, respectively, with "IMP" reflecting the imputation nature of the latter algorithm. The estimator from the $L_2$Boost-IMP algorithm is denoted as $\hat{f}_n^{\mathrm{IMP}}$.

These two algorithms differ in their approach to addressing interval-censored data. The $L_2$Boost-CUT method adjusts the loss function so its expectation recovers that of the original $L_2$ loss $L$, as required in (6), whereas the $L_2$Boost-IMP method preserves the functional form of the original loss $L$ but replaces its first argument with the transformed response $\tilde{Y}_1(\mathcal{O}_i)$ in (8). Therefore, the loss functions for those two algorithms are distinct:

$$L_{\mathrm{CUT}}(\mathcal{O}_i, f(X_i)) \neq L(\tilde{Y}_1(\mathcal{O}_i), f(X_i)).$$

The risk from $L_2$Boost-CUT satisfies Proposition 1 (proved in Appendix D), but this property does not hold for $L_2$Boost-IMP. Nevertheless, due to the linear derivative of the $L_2$ loss in its first argument, the following connection emerges:

$$\partial \hat{L}\left(\mathcal{O}_i, f^{(t-1)}(X_i)\right) = \partial L\left(\hat{Y}_1(X_i), f^{(t-1)}(X_i)\right) = \hat{Y}_1(X_i) - f^{(t-1)}(X_i). \tag{11}$$

This leads to closely related increment terms in both methods, and as such, $L_2$Boost-CUT and $L_2$Boost-IMP mainly differ in the stopping criterion, suggesting that they often yield similar results, as observed in the experiment results in Section 5 and Appendix G. Further discussions on these two methods are provided in Appendix E.3.

## 3.3 BASE LEARNERS AND SURVIVOR FUNCTION

To outline the key steps in Algorithm 1, we begin with notation related to the base learners at each iteration. For iteration $t = 1, 2, \ldots$, let $\vec{h}^{(t)} = \left(\hat{h}^{(t)}(X_1), \ldots, \hat{h}^{(t)}(X_n)\right)^{\top}$, where $\hat{h}^{(t)}$ is the base learner at iteration $t$, defined in Line 4 of Algorithm 1. For $k = 1, 2$, let $\vec{Y}_k = \left(\hat{Y}_k(\mathcal{O}_1), \ldots, \hat{Y}_k(\mathcal{O}_n)\right)^{\top}$, where $\hat{Y}_k(\mathcal{O}_i)$ represents an estimate for (8). For $f^{(t-1)}$ in Line 5 of Algorithm 1, we define $\vec{f}^{(t-1)} = \left(f^{(t-1)}(X_1), \ldots, f^{(t-1)}(X_n)\right)^{\top}$, and compute the residual:

$$\vec{u}^{(t-1)} = \vec{Y}_1 - \vec{f}^{(t-1)}. \tag{13}$$

Algorithm 1 iteratively updates the base learners that map $\mathcal{X}$ to $\mathcal{Y}$ for each iteration. In our implementation, we use *linear smoothers* (Buja et al., 1989) and focus particularly on *smoothing splines*, as in Bühlmann & Yu (2003). Linear smoothers are versatile and cover a wide range of function classes, including least squares, regression splines, kernels, and many others.

At each iteration, the residual $\vec{u}^{(t-1)}$ is smoothed using a *smoother matrix*, represented by an $n \times n$ matrix $\Psi$, which transforms the residual into an updated base learner:

$$\vec{h}^{(t)} = \Psi \vec{u}^{(t-1)}. \tag{14}$$

---

**Algorithm 1** $L_2$Boost-CUT

---

1: Take $f^{(0)} = \arg\min_h \left[ n^{-1} \sum_{i=1}^n \left\{ \hat{Y}_1(\mathcal{O}_i) - h(X_i) \right\}^2 \right]$ and set $\eta = n^{-w}$ for a given $w \geq 1$;

2: **for** iteration $t$ with $t = 1, 2, \ldots$ **do**

3:     (i) calculate $\partial \hat{L}\left(\mathcal{O}_i, f^{(t-1)}(X_i)\right) \triangleq \left. \frac{\partial \hat{L}(u,v)}{\partial v} \right|_{u=\mathcal{O}_i, v=f^{(t-1)}(X_i)}$ for $i = 1, \ldots, n$;

4:     (ii) find $\hat{h}^{(t)} = \arg\min_{h^{(t)}} \left[ n^{-1} \sum_{i=1}^n \left\{ -\partial \hat{L}\left(\mathcal{O}_i, f^{(t-1)}(X_i)\right) - h^{(t)}(X_i) \right\}^2 \right]$;

5:     (iii) for regression tasks, update $f^{(t)}(X_i)$ as (3) for $i = 1, \ldots, n$;

6:        for classification tasks, update $f^{(t)}(X_i)$ as (5) for $i = 1, \ldots, n$;

7:     **if** at iteration $\tilde{t}$,

$$\left| n^{-1} \sum_{i=1}^n \hat{L}\left(\mathcal{O}_i, f^{(\tilde{t})}(X_i)\right) - n^{-1} \sum_{i=1}^n \hat{L}\left(\mathcal{O}_i, f^{(\tilde{t}-1)}(X_i)\right) \right| \leq \eta \qquad (12)$$

8:        **then** stop iteration and define the final estimator as $\hat{f}_n^{\text{CUT}} = f^{(\tilde{t}-1)}$.

9:     **end if**

10: **end for**

---

Here, $\Psi$ is determined by the chosen linear smoother, which may depend on features but not on $\vec{u}^{(t-1)}$ (Hastie et al., 2009, Chapter 5.4.1). We provide further details on smoothing splines in Appendix B.

The execution of Algorithm 1 requires calculations of $\tilde{Y}_k(\mathcal{O}_i)$ in (8), which hinges on consistently estimating the conditional survivor function $S(y|X_i)$; here $S(y|X_i)$ is interpreted as $S(y|X_i = x_i)$ for any realization $x_i$ of $X_i$; similar considerations apply for functions of $X_i$ or conditioning on $X_i$ throughout the paper, with $X_i$ evaluated at its realization. While an estimator of $S(y|X_i)$ with a faster convergence rate yields a more efficient estimator $\hat{f}_n^{\text{CUT}}$, consistency suffices to ensure the validity of our methods. Instead of pursuing faster convergence through parametric approaches, which are vulnerable to model misspecification, we prioritize robustness by opting for the *interval censored recursive forests* (ICRF) method (Cho et al., 2022), whose consistency has been established by Cho et al. (2022). ICRF is a tree-based, nonparametric method designed for estimating survivor functions for interval-censored data. It serves as a component within our framework for developing boosting methods for regression and classification with interval-censored data, aiming to predict a transformed target variable $g(Y)$ described in Section 2. Further details on this estimation are provided in Appendix C.

## 4 THEORETICAL RESULTS

Assuming consistent estimation of $S(y|X_i)$, we now develop theoretical guarantees for the proposed methods, both in regression and classification contexts, with the proofs deferred to Appendix D.

### 4.1 REGRESSION

Consider the regression model

$$g(Y_i) = \phi(X_i) + \epsilon_i \quad \text{for } i = 1, \ldots, n, \qquad (15)$$

where $\epsilon_i$ are independent and identically distributed with $E(\epsilon_i) = 0$ and $\text{var}(\epsilon_i) = \sigma^2 < \infty$, $\phi$ is an unknown smooth function that can be linear or nonlinear, and $g$ is a user-specified transformation, as discussed in Section 2.

At iteration $t = 1, 2, \ldots$, the $L_2$Boost-CUT and $L_2$Boost-IMP methods map the interval-censored data $\mathcal{O}^{\text{IC}}$ (described in Section 3.1) to $\vec{f}^{(t)}$, following Algorithm 1. For $k = 1, 2$, we first utilize (8) and ICRF to construct $\vec{Y}_k$ from $\mathcal{O}^{\text{IC}}$, then apply the conventional $L_2$Boost method (described in Section 2) to $\left\{ \left\{ X_i, \hat{Y}_1(\mathcal{O}_i) \right\} : i = 1, \ldots, n \right\}$. Specifically, for $\vec{Y}_1$ defined in Section 3.3, these

procedures can be formulated as:

$$\vec{f}^{(t)} = B^{(t)}\vec{Y}_1, \tag{16}$$

where $B^{(t)}$ represents an $n \times n$ matrix that transforms $\vec{Y}_1$ to $\vec{f}^{(t)}$ at each given $t$. The following proposition shows that $B^{(t)}$ can be represented in terms of the smoother matrix $\Psi$.

**Proposition 2.** *For $t = 1, 2, \ldots$, let $B^{(t)}$ denote the $L_2$Boost-CUT or $L_2$Boost-IMP operator at iteration $t$. Let $\Psi$ represent the smoother matrix for the chosen linear smoother. Then, $B^{(t)} \triangleq I - (I - \Psi)^{t+1}$ for $t = 1, 2, \ldots$, where $I$ is the $n \times n$ identity matrix.*

Next, we examine the averaged mean squared error (MSE) for using $f^{(t)}$ (defined in Line 5 of Algorithm 1) to predict $\phi$ in (15), similar to Bühlmann & Yu (2003). The MSE is defined as

$$\text{MSE}(t, \Psi; \phi) = n^{-1} \sum_{i=1}^{n} E\left[\left\{f^{(t)}(X_i) - \phi(X_i)\right\}^2\right], \tag{17}$$

where $\text{MSE}(t, \Psi; \phi)$ depends on $\Psi$ via (16), and the expectation is taken with respect to the joint distribution for the random variables in $\mathcal{O}^{\text{IC}}$ defined in Section 3.1. Here, $\phi(X_i)$ is treated as a constant for each realization of $X_i$. Let

$$\text{var}(t, \Psi) \triangleq n^{-1} \sum_{i=1}^{n} \text{var}\left\{f^{(t)}(X_i)\right\} \text{ and bias}^2(t, \Psi; \phi) \triangleq n^{-1} \sum_{i=1}^{n} \left[E\left\{f^{(t)}(X_i)\right\} - \phi(X_i)\right]^2 \tag{18}$$

denote the averaged variance and the averaged squared bias for using $f^{(t)}$ to predict $\phi$, respectively.

**Proposition 3.** *MSE$(t, \Psi; \phi)$ in (17) can be decomposed into the sum of var$(t, \Psi)$ and bias$^2(t, \Psi; \phi)$ in (18):*

$$MSE(t, \Psi; \phi) = var(t, \Psi) + bias^2(t, \Psi; \phi).$$

Let $\vec{\phi}$ denote the vector $(\phi(X_1), \ldots, \phi(X_n))^\top$. Assume that the smoother matrix $\Psi$ is real, symmetric, and has eigenvalues $\{\lambda_1, \ldots, \lambda_n\}$ with corresponding normalized eigenvectors $\{Q_1, \ldots, Q_n\}$. Let $Q$ denote the matrix with $Q_l$ being the $l$th column for $l = 1, \ldots, n$, and let $\mu = (\mu_1, \ldots, \mu_n)^\top \triangleq Q^\top\vec{\phi}$ be the function vector in the linear space spanned by the eigenvectors of $\Psi$. Let $\mathcal{O}$ be a collection of random variables, drawn from the same distributions as the elements of $\mathcal{O}_i$, and let $\hat{\sigma}^2 = \text{var}\left\{\hat{Y}_1(\mathcal{O})\right\}$.

**Proposition 4.** *Assume regularity condition (C1) in Appendix A. Then var$(t, \Psi)$ and bias$^2(t, \Psi; \phi)$ in (18) can be, respectively, simplified as*

$$var(t, \Psi) = \hat{\sigma}^2 n^{-1} \sum_{l=1}^{n} \left\{1 - (1 - \lambda_l)^{t+1}\right\}^2 \text{ and bias}^2(t, \Psi; \phi) = n^{-1} \sum_{l=1}^{n} \mu_l^2 (1 - \lambda_l)^{2t+2}.$$

These results align with Proposition 3 in Bühlmann & Yu (2003), and they suggest that the iteration index $t$ can be interpreted as a "smoothing parameter" that balances the bias–variance trade-offs.

**Corollary 1.** *Assume the regularity condition in Proposition 4. If $\lambda_l \in \{0, 1\}$ for $l = 1, \ldots, n$, then $B^{(t)} = \Psi$ for $t = 1, 2 \ldots$.*

This corollary implies that in special cases, such as when the smoother matrix has eigenvalues of 0 or 1 (e.g., projection smoothers (Hastie et al., 2009, Chapter 5.4), like least squares, polynomial regression, and regression splines (Buja et al., 1989)), the $L_2$Boost-CUT and $L_2$Boost-IMP algorithms cease to provide additional boosting to learners.

**Proposition 5.** *Assume the regularity condition in Proposition 4 and condition (C2) in Appendix A. Then, as the number of boosting iterations $t$ increases, bias$^2(t, \Psi; \phi)$ decays exponentially and var$(t, \Psi)$ exhibits an exponential increase, yielding*

$$\lim_{t \to \infty} MSE(t, \Psi; \phi) = \hat{\sigma}^2.$$

Similar to Theorem 1(a) of Bühlmann & Yu (2003), this proposition implies that running the $L_2$Boost-CUT and $L_2$Boost-IMP algorithms infinitely is generally not beneficial: the MSE will not decrease below $\hat{\sigma}^2$, and excessive boosting may lead to overfitting.

**Proposition 6.** *Assume the regularity conditions in Proposition 5 and condition (C3) in Appendix A. Then there exists a positive integer $t_0$, such that $MSE(t_0, \Psi; \phi)$ is strictly smaller than $\hat{\sigma}^2$.*

This result, complementary to Theorem 1(b) of Bühlmann & Yu (2003), shows that in contrast to condition (C2), when a stronger condition (C3) holds, the $L_2$Boost-CUT and $L_2$Boost-IMP algorithms can achieve an MSE smaller than $\hat{\sigma}^2$, even with a finite number of iterations.

**Theorem 1.** *Assume the regularity conditions in Proposition 6 and condition (C4) in Appendix A. Then for $m_0 \geq 2$ in condition (C4), the first $\lfloor m_0 \rfloor$ iterations of the $L_2$Boost-CUT algorithm (i.e., Algorithm 1) improve the MSE over the unboosted base learner algorithm (i.e., linear smoothers), where $\lfloor \cdot \rfloor$ is the floor function.*

Condition (C4) basically requires base learners to be weak (see Appendix A for details). This theorem suggests that the $L_2$Boost-CUT and $L_2$Boost-IMP algorithms consistently outperform an unboosted weak learner. This result complements Theorem 1(c) in Bühlmann & Yu (2003).

**Theorem 2.** *Let $\hat{\epsilon}_i \triangleq \hat{Y}_1(\mathcal{O}_i) - E\left\{\hat{Y}_1(\mathcal{O}_i)\right\}$. Assume the regularity conditions in Proposition 5 and condition (C5) in Appendix A. Then for a positive constant $q$, there exists a positive constant $C$ that is functionally independent of $t$ (but may be dependent on $q$ and $n$) such that as $t \to \infty$,*

$$n^{-1} \sum_{i=1}^{n} E\left[\left\{f^{(t)}(X_i) - \phi(X_i)\right\}^q\right] = E\left(\hat{\epsilon}_i^q\right) + O(\exp(-Ct)). \tag{19}$$

For $q = 2$, Theorem 2 directly yields Proposition 5. In the following development, we may write the iteration index $t$ as $t_n$ to stress its dependence on the sample size $n$.

**Theorem 3.** *Consider the model (15) with $X_i \in \mathbb{R}$ satisfying Assumption (A) of Bühlmann & Yu (2003). If base learner $\hat{h}^{(t)}$ is the smoothing spline learner of smoothing degree $r$ and degrees of freedom $df$, as detailed in (B.1) and (B.2) in Appendix B, $\phi \in \mathcal{W}^{(v,2)}(\mathcal{X})$ with $v \geq r$, then there exists an optimal number of iterations $t_n = O\left(n^{2r/(2v+1)}\right)$ such that $f^{(t_n)}$ achieves the minimax-optimal rate, $O\left(n^{-2v/(2v+1)}\right)$, for the function class $\mathcal{W}^{(v,2)}(\mathcal{X})$ in terms of MSE, as defined in (17).*

Theorem 3 shows that the $L_2$Boost-CUT and $L_2$Boost-IMP algorithms achieve minimax optimality with a smoothing spline learner for one-dimensional feature $X_i$. Even if the base learner has smoothness order $r < v$, the algorithms still adapt to higher-order smoothness $v$, attaining the optimal MSE rate $O\left(n^{-2v/(2v+1)}\right)$ asymptotically, similar to $L_2$Boost in Bühlmann & Yu (2003). When paired with a smoothing spline learner, the $L_2$Boost-CUT and $L_2$Boost-IMP algorithms can adapt to any $v$th-order smoothness of $\mathcal{W}^{(v,2)}(\mathcal{X})$. For example, with a cubic smoothing spline ($r = 2$) and $v = 2$, $f^{(t_n)}$ can achieve the optimal MSE rate of $n^{-4/5}$ by selecting $t_n = O\left(n^{4/5}\right)$. For higher-order smoothness, such as $v$ exceeding $r$ with $v = 3$, $f^{(t_n)}$ can attain an optimal MSE rate of $n^{-6/7}$ with $t_n = O\left(n^{4/7}\right)$. While the $L_2$Boost-CUT and $L_2$Boost-IMP algorithms can also adapt to functions with lower-order smoothness ($v < r$), this adaptability may not provide additional gains in such scenarios, as noted by Bühlmann & Yu (2003) for the scenario with complete data.

## 4.2 CLASSIFICATION

In classification tasks, the goal is to estimate the probability $P(Y_i > s)$ for given time $s$ in order to determine a predicted value for new features. In this instance, we define $g(Y_i) = 2I(Y_i > s) - 1$, and let $g_s(Y_i)$ denote it to stress the dependence on $s$. At iteration $t$, $L_2$Boost-CUT provides an estimate, denoted $f_s^{(t)}$, for $E\{g_s(Y_i)|X_i\} = 2p_s(X_i) - 1$, where $p_s(X_i) \triangleq E\{I(Y_i > s)|X_i\}$. Here, $f_s^{(t)}$ represents $f^{(t)}$ in Algorithm 1 with the dependence on $s$ explicitly spelled out.

In line with Bühlmann & Yu (2003), estimating $2p_s(X_i) - 1$ can be loosely regarded as analogous to (15):

$$g_s(Y_i) = 2p_s(X_i) - 1 + \epsilon_i \quad \text{for } i = 1, \ldots, n,$$

where the noise term $\epsilon_i$ has $E(\epsilon_i) = 0$ and $\text{var}(\epsilon_i) = 4p_s(X_i)\{1 - p_s(X_i)\}$. Because the variances $\text{var}(\epsilon_i)$ for $i = 1, \ldots, n$ are upper bounded by 1, Theorem 3 can be modified to give the optimal MSE rates for using the $L_2$Boost-CUT and $L_2$Boost-IMP methods to estimate $p_s$.

**Theorem 4.** *Consider the setup in Theorem 3. If $p_s$ belongs to $\mathcal{W}^{(v,2)}(\mathcal{X})$, then there exists $t_n = O\left(n^{2r/(2v+1)}\right)$ such that $f^{(t_n)}$ achieves the minimax-optimal rate, $O\left(n^{-2v/(2v+1)}\right)$, which minimizes MSE as defined in (17).*

Next, similar to Bühlmann & Yu (2003), we define the averaged Bayes risk (BR) for any $s > 0$:

$$\mathrm{BR}_s = n^{-1} \sum_{i=1}^{n} \Pr\left\{\mathrm{sign}\left(2p_s(X_i) - 1\right) \neq g_s(Y_i)\right\}.$$

**Theorem 5.** *Assume the regularity conditions in Theorem 4 hold. Then there exists $t_n = O\left(n^{2r/(2v+1)}\right)$ such that*

$$n^{-1} \sum_{i=1}^{n} \Pr\left(f_s^{(t_n)}(X_i) \neq g_s(Y_i)\right) - BR_s = O\left(n^{-v/(2v+1)}\right).$$

Theorem 5 shows that, for $L_2$Boost-CUT and $L_2$Boost-IMP, the difference between the empirical misclassification rate and $\mathrm{BR}_s$ is of order $O\left(n^{-v/(2v+1)}\right)$, which approaches 0 as $n \to \infty$.

## 5 EXPERIMENTS AND DATA ANALYSES

**Experimental setup.** Each experimental setup involves conducting 300 experiments with a sample size $n$. For $i = 1, \ldots, n$, let $X_i = (X_{1,i}, \ldots, X_{p,i})^\top$, where the $X_{l,i}$ are independently drawn from the uniform distribution over $[0,1]$ for $l = 1, \ldots, p$ and $i = 1, \ldots, n$. The responses $Y_i$ are then independently generated from an accelerated failure time (AFT) model (Sun, 2006), given by (15), where $g(u) = \log u$, and the error terms $\epsilon_i$ are independently generated from either a normal distribution $N(0, \sigma^2)$ with variance $\sigma^2$ or the logistic distribution with location and scale parameters set as 0 and $1/8$, respectively. For $i = 1, \ldots, n$, we generate $m$ monitoring times independently from a uniform distribution over $[0, \tau]$, and then order them as $u_{i,1} < u_{i,2} < \ldots < u_{i,m}$. We set $n = 500$, $\sigma = 0.25$, $p = 1$, $\tau = 6$, $m = 3$, and $\phi(X_i) = \beta_0|X_i - 0.5| + \beta_1 X_i^3 + \beta_2 \sin(\pi X_i)$, with $\beta_0 = 1$, $\beta_1 = 0.8$, and $\beta_2 = 0.8$.

**Learning methods and evaluation metrics.** We analyze synthetic data using the proposed $L_2$Boost-CUT (CUT) and $L_2$Boost-IMP (IMP) methods, as opposed to three other methods: the oracle (O) method uses the oracle dataset $\mathcal{O}_O^{\mathrm{TR}} \triangleq \{\{\phi(X_i), X_i\} : i = 1, \ldots, n_1\}$ with true values of $\phi(X_i)$, the reference (R) method uses the complete dataset $\mathcal{O}_C^{\mathrm{TR}} \triangleq \{\{Y_i, X_i\} : i = 1, \ldots, n_1\}$, and the naive (N) method employs a surrogate response $\tilde{Y}_i \triangleq \frac{1}{2}(L_i + R_i)$ if $R_i < \infty$ and $\tilde{Y}_i \triangleq L_i$ otherwise, together with $X_i$. While the O and R methods require full data availability, which is unrealistic in real-world applications, they provide upper performance bounds under ideal, fully informed conditions. This, in turn, benchmarks how our methods perform in realistic settings.

Synthetic data are split into training and test datasets in a $4 : 1$ ratio. We assess the performance of each method using sample-based maximum absolute error (SMaxAE), sample-based mean squared error (SMSqE), and sample-based Kendall's $\tau$ (SKDT), for regression tasks, along with *sensitivity* and *specificity* for classification tasks. Details are provided in Appendix F.1.

**Experiment results.** Figure 1 summarizes the SMaxAE, SMSqE, and SKDT values using boxplots for predicting survival times. The N method produces the largest SMaxAE and SMSqE values yet the smallest SKDT values, whereas the proposed CUT and IMP methods outperform the N method, yielding values fairly comparable to those of the R method. Figure 2 displays the sensitivity and specificity metrics for predicting survival status, where sensitivity plots for $s = 4$ and specificity plots for $s = 1$ are omitted because no corresponding positive and negative cases exist; and the CUT and IMP methods produce identical lines. The N method produces similar specificity values but significantly lower sensitivity values compared to the proposed CUT and IMP methods. To evaluate how the performance of the proposed methods is affected by various factors, including sample sizes, data generation models, noise levels, and different implementation ways of ICRF, we conduct additional experiments in Appendix G.

**Data analyses.** We apply the proposed CUT and IMP methods as well as the N method to analyze two datasets, Signal Tandmobiel® data and Bangkok HIV data, whose details are included in Appendix F.4. In addition, we implement a procedure (denoted COX) based on the Cox model, though

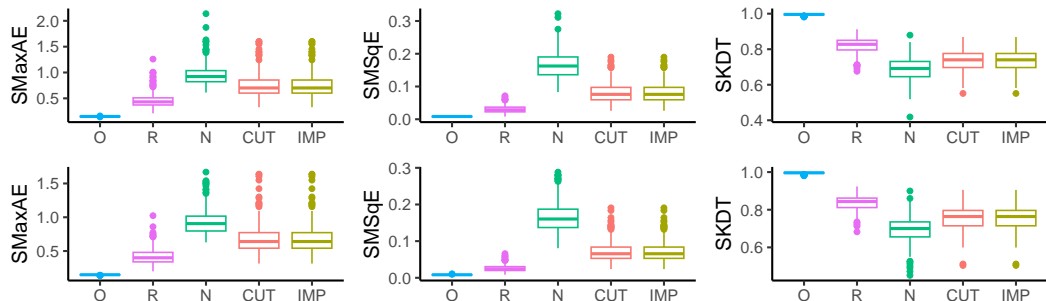

Figure 1: *Experiment results of predicting survival times. The top and bottom rows correspond to the lognormal AFT and loglogistic AFT models, respectively.*

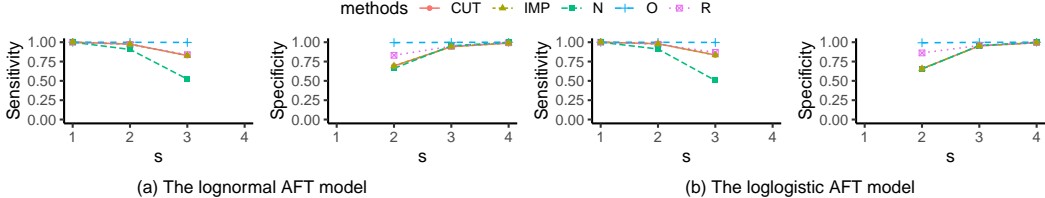

Figure 2: *Experiment results of predicting survival status.*

its results are not directly comparable to those three methods. Details are provided in Appendix G.2. Figure 3 reports boxplots for the values of $\exp\left\{\hat{f}^*(X_i)\right\}$, where $\hat{f}^*(X_i)$ represents an estimate from a method. Clearly, both the CUT and IMP methods yield comparable estimates, while the N method produces smaller estimates. The results from COX appear to be closer to those from the two proposed methods than those from the N method.

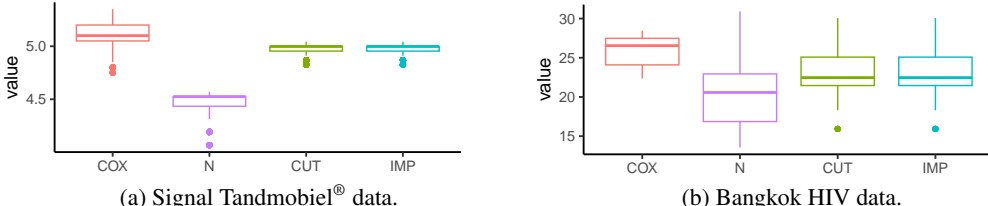

Figure 3: *Boxplots of data analysis results.*

## 6    DISCUSSION

In this paper, we introduce the boosting algorithms tailored for interval-censored data. These methods offer the flexibility in predicting various survival outcomes, including survival times, survival probabilities, and survival status at specified time points. Further discussions, including computational complexity and extensions, are included in Appendix E. Like all methods, the validity of our approaches depends on certain conditions, as outlined in Appendix A. For example, using smoother matrices that fail to meet these conditions may compromise their effectiveness. Condition (C4) states the importance of employing weak learners as base learners. Using overly strong base learners could violate this assumption and negatively impact the performance. Our methods inherently depend on estimation of the survivor function, which typically utilizes ICRF. While this ensures robust and consistent estimation, it requires additional computational cost as shown in Appendix E.2 and Table F.1 in Appendix F.3. This computational cost reflects the price paid to achieve the methodological robustness that our approaches offer.

## ACKNOWLEDGMENTS

This research was supported by the Natural Sciences and Engineering Research Council of Canada (NSERC). Yi is a Canada Research Chair in Data Science (Tier 1). Her research was also supported by the Canada Research Chairs Program.

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

APPENDICES: TECHNICAL DETAILS AND ADDITIONAL EXPERIMENT RESULTS

## A    REGULARITY CONDITIONS

For features, we consider the same assumptions as in Bühlmann & Yu (2003). Furthermore, we assume the following conditions.

(C1) The smoother matrix $\Psi$ is real and symmetric having eigenvalues $\{\lambda_1, \ldots, \lambda_n\}$ and corresponding normalized eigenvectors $\{Q_1, \ldots, Q_n\}$, with $Q_i^\top Q_i = 1$ for $i = 1, \ldots, n$.

(C2) The eigenvalues $\lambda_k$ satisfy $0 < \lambda_k \leq 1$ for $k = 1, \ldots, n$.

(C3) There exists at least one $k$ in $\{1, \ldots, n\}$ such that $\lambda_k < 1$.

(C4) There exists $m_0 \geq 2$ such that for all $k$ with $\lambda_k < 1$,

$$\mu_k^2/\hat{\sigma}^2 > 1/(1 - \lambda_k)^{m_0} - 1.$$

(C5) For $\hat{\epsilon}$ defined in Theorem 2, $E\left(\hat{\epsilon}^q\right) < \infty$ for $q = 1, 2, \ldots$.

The smoother matrix $\Psi$, which satisfies conditions (C1) and (C2), includes projection smoothers and shrinking smoothers (Hastie et al., 2009, Chapter 5.4), such as smoothing splines introduced in Appendix B. To clarify further, shrinking smoothers also satisfy condition (C3), whereas projection smoothers do not. Condition (C4) encompasses the condition in Theorem 1(c) of Bühlmann & Yu (2003) as a special case. It can be interpreted as follows: a large value on the left-hand side suggests that $\phi$ is relatively complex compared to the estimated noise level $\hat{\sigma}^2$, while a small value on the right-hand side implies that $\lambda_k$ is small, which indicates that the learner either applies strong shrinkage or smoothing in the $k$th eigenvector direction, or is inherently weak in that direction.

## B    SMOOTHING SPLINES

To introduce smoothing splines, we start with considering the simple case where the features $X_i$ is one-dimensional. For $\mathcal{X} \subseteq \mathbb{R}$ and a positive $v$, let

$$\mathcal{W}^{(v,2)}(\mathcal{X}) = \left\{ g : \mathcal{X} \to \mathbb{R} \,\middle|\, g \text{ is differentiable up to order } v \text{ and } \int_{x \in \mathcal{X}} \left\{ g^{(v)}(x) \right\}^2 dx < \infty \right\}$$

denote the *Sobolev space* of the $v$th-order smoothed functions defined over $\mathcal{X}$.

Let $r$ be a positive integer. At iteration $t$ in Algorithm 1, we find a smoothing spline learner of degree $r$, denoted $\hat{h}^{(t)}$, by solving the penalized least squares problem:

$$\hat{h}^{(t)} = \underset{h^{(t)} \in \mathcal{W}^{(v,2)}(\mathcal{X})}{\arg\min} \left[ n^{-1} \sum_{i=1}^{n} \left\{ -\partial \hat{L}\left( \mathcal{O}_i, f^{(t-1)}(X_i) \right) - h^{(t)}(X_i) \right\}^2 + \lambda \int_{x \in \mathcal{X}} \left\{ h^{(t)(r)}(x) \right\}^2 dx \right],$$
(B.1)

where $h^{(t)(r)}$ represents the $r$th order derivative of $h^{(t)}$, and $\lambda$ is a tuning parameter. Here, the dependence of $\hat{h}^{(t)}$ on the tuning parameter $\lambda$, smoothness degree $r$, and $v$ is suppressed in the notation.

Taking $v = r = 2$ often offers a viable way to handle practical problems, yielding cubic smoothing splines (Hastie et al., 2009). Varying $\lambda$ from 0 to $\infty$ accommodates different forms of $\hat{h}^{(t)}$. Setting $\lambda = 0$ imposes no penalty in (B.1), and $\hat{h}^{(t)}$ is then a natural spline that interpolates $\left( X_i, -\partial \hat{L}\left( \mathcal{O}_i, f^{(t-1)}(X_i) \right) \right)$ for $i = 1, \ldots, n$; taking $\lambda = \infty$ leads $\hat{h}^{(t)}$ in (B.1) to be the $r$th order polynomials if $v \geq r$ (Wang, 2011). The larger $\lambda$ is, the weaker a base learner is. Though the value of $\lambda$ is crucial to the success of the learning process, it is difficult to decide an optimal or reasonable value for $\lambda$ when using smoothing splines. Alternatively, noting that a more interpretable parameter, *degrees of freedom*, defined as

$$df \triangleq \text{Trace}(\Psi),$$
(B.2)

is monotone in $\lambda$, Hastie et al. (2009, p.158) suggested to invert the relationship and specify $\lambda$ by fixing $df$.

Though we start with the infinite dimensional space $\mathcal{W}^{(v,2)}(\mathcal{X})$, $\hat{h}^{(t)}$ in (B.1) is showed to be a natural polynomial spline with knots at all distinct $X_i$ for $i = 1, \ldots, n$ (Eubank, 1988), which belongs to a finite dimensional space (Hastie et al., 2009, Chapter 5.4; Wang, 2011). Let $\{N_l : l = 1, \ldots, n\}$ denote a set of $n$ second-order differentiable basis functions for the family of natural splines, and let $N$ and $\Omega$ denote matrices with the $(i, l)$ entry equaling $N_l(X_i)$ and $\int_{x \in \mathcal{X}} N_i''(x) N_l''(x) dx$, respectively. Let $\hat{\theta}_l$ denote the $l$th element of $\left(N^\top N + \lambda \Omega\right)^{-1} N^\top \vec{u}^{(t-1)}$. Further, assuming the $X_i$ are all distinct for $i = 1, \ldots, n$, Hastie et al. (2009, Chapter 5.4) showed that $\hat{h}^{(t)}$ in (B.1) with $v = r = 2$ can be written as

$$\hat{h}^{(t)} = \sum_{l=1}^n N_l \hat{\theta}_l.$$

That is, the cubic smoothing spline with a pre-specified $\lambda$ is a linear smoother with $\Psi$ in (14) equaling $N \left(N^\top N + \lambda \Omega\right)^{-1} N^\top$ (Hastie et al., 2009).

Next, we consider the general case where $\mathcal{X} \subseteq \mathbb{R}^p$, for which we may employ (B.1) elementwisely to update a base learner in a manner similar to Bühlmann & Yu (2003). Specifically, at iteration $t$, consider each component $X_{l,i}$ of $X_i \triangleq (X_{1,i}, \ldots, X_{p,i})^\top$, we employ the smoothing spline with the selected feature $X_{\hat{l}_t, i}$, where $\hat{l}_t \in \{1, \ldots, p\}$ is determined by

$$\hat{l}_t = \arg\min_{1 \le l \le p} \sum_{i=1}^n \left\{ -\partial \hat{L}\left(\mathcal{O}_i, f^{(t-1)}(X_i)\right) - \hat{h}_l^{(t)}(X_{l,i}) \right\}^2.$$

Here, $\hat{h}_l^{(t)}(X_{l,i})$ is the smoothing spline as in (B.1) obtained from replacing $X_i$ in (B.1) with the feature $X_{l,i}$.

## C  ESTIMATION WITH INTERVAL-CENSORED RECURSIVE FORESTS

Here, we describe our estimation detail with the interval-censored recursive forests algorithm. Let $T$ and $D$ denote the total number of iteration and the number of bootstrap samples, respectively. We describe the estimation procedure as follows.

Step 1. We set an initial estimate for $S(y|X_i)$, denoted $\hat{S}^{(0)}(y|X_i)$. A simple way is to set $\hat{S}^{(0)}(y|X_i)$ to be the nonparametric maximum likelihood estimate (NPMLE) of unconditional survivor function of $Y_i$, denoted $\hat{S}(y)$ (Turnbull, 1976). Then for $i = 1, \ldots, n$, we employ the kernel smoothing technique to obtain a smoothed estimate of $S(y|X_i)$, denoted by $\tilde{\lambda}^{(0)}(y|X_i)$. That is,

$$\tilde{\lambda}^{(0)}(y|X_i) = 1 + \int_0^y \int_{\mathbb{R}^+} \frac{1}{h} K_h(s - v) d\hat{S}^{(0)}(v|X_i) ds,$$

where $K_h$ is a kernel function with bandwidth $h > 0$.

Step 2. At iteration $t$, we draw $D$ independent bootstrap samples with size $\lceil 0.95n \rceil$ from $\mathcal{O}^{\text{IC}}$, denoted as $\mathcal{O}_1^{(t)}, \ldots, \mathcal{O}_D^{(t)}$, where $\lceil \cdot \rceil$ is the ceiling function; and keep $\mathcal{O}^{\text{IC}} \backslash \mathcal{O}_d^{(t)}$ as the out-of-bag sample for $d = 1, \ldots, D$, denoting them as $\mathcal{O}_1^{\text{OOB},(t)}, \ldots, \mathcal{O}_D^{\text{OOB},(t)}$. For each bootstrap sample $\mathcal{O}_d^{(t)}$ with $d = 1, \ldots, D$, we build a tree using two-sample testing rules for interval-censored data based on the conditional survivor function $\hat{S}_d^{(t-1)}(y|X_i)$, say the generalized Wilcoxon's rank sum (GWRS) test or the generalized logrank (GLR) test (Cho et al., 2022).

Specifically, at each node, we randomly pick $\lceil \sqrt{p} \rceil$ features, with $p$ denoting the dimension of features, and then we find the optimal cutoff suggested by GWRS or GLR. Let $L_d^{(t)}$ denote the total number of terminal nodes of the resulting tree for the $d$th bootstrap sample at iteration $t$. For $l = 1, \ldots, L_d^{(t)}$, let $A_{d,l}^{(t)}$ denote the $l$th terminal node in the $d$th tree. At the

$l$th terminal node of the tree, we estimate the survival probabilities for each node, denoted $\hat{S}_{d,l}^{(t)}\left(y|A_{d,l}^{(t)}\right)$, using the *quasi-honest* or *exploitative* approaches. The quasi-honesty approach employs the NPMLE based on raw interval-censored data, whereas the exploitative approach averages the estimates of the conditional survivor function from iteration $t-1$ (Cho et al., 2022). The exploitative approach is computationally efficient, while the estimator obtained from the quasi-honesty approach exhibits uniform consistency, provided regularity conditions (Cho et al., 2022). However, the finite sample performance of these two approaches varies, with neither consistently outperforming the other (Cho et al., 2022).

To presume some degree of smoothness in the true survivor function, $\hat{S}_{d,l}^{(t)}$ is further smoothed as $\tilde{\lambda}_{d,l}^{(t)}$ using the kernel-smoothing technique, yielding a smoothed estimate of the conditional survivor function $\tilde{\lambda}(y|X_i)$ (Cho et al., 2022).

Step 3. Calculate the conditional survivor function for the $d$th tree and its smoothed version as

$$\hat{S}_d^{(t)}(y|X_i) = \sum_{l=1}^{L_d^{(t)}} \hat{S}_{d,l}^{(t)}\left(y|A_{d,l}^{(t)}\right) I\left(X_i \in A_{d,l}^{(t)}\right)$$

and

$$\tilde{\lambda}_d^{(t)}(y|X_i) = \sum_{l=1}^{L_d^{(t)}} \tilde{\lambda}_{d,l}^{(t)}\left(y|A_{d,l}^{(t)}\right) I\left(X_i \in A_{d,l}^{(t)}\right).$$

Calculate the out-of-bag error as the integrated mean squared error (IMSE)

$$\epsilon_d^{(t)} \triangleq \frac{1}{n^{\text{OOB}}} \sum_{i=1}^{n^{\text{OOB}}} \frac{1}{\tau - (R_i \wedge \tau) + (L_i \wedge \tau)}$$
$$\times \left\{ \int_0^{L_i \wedge \tau} \left(1 - \tilde{\lambda}_d^{(t)}(s|X_i)\right)^2 ds + \int_{R_i}^{R_i \wedge \tau} \left(\tilde{\lambda}_d^{(t)}(s|X_i)\right)^2 ds \right\},$$

where $n^{\text{OOB}} \triangleq n - \lceil 0.95n \rceil$ denotes the sample size of $\mathcal{O}_d^{\text{OOB},(t)}$, and $a \wedge b \triangleq \min(a,b)$.

Step 4. Averaging the corresponding quantities over $D$ trees, we obtain that

$$\hat{S}^{(t)}(y|X_i) = \frac{1}{D}\sum_{d=1}^D \hat{S}_d^{(t)}(y|X_i),\ \tilde{\lambda}^{(t)}(y|X_i) = \frac{1}{D}\sum_{d=1}^D \tilde{\lambda}_d^{(k)}(y|X_i),\ \text{and } \epsilon^{(t)} = \frac{1}{D}\sum_{d=1}^D \epsilon_d^{(t)}.$$

Then the final estimate of $S(y|X_i)$ is determined as $\tilde{\lambda}(y|X_i) = \tilde{\lambda}_d^{(t_{\text{opt}})}(y|X_i)$, with $k_{\text{opt}} = \arg\min_{1 \le t \le T} \epsilon^{(t)}$.

Step 5. We approximate $\int_{u_{j-1}}^{u_j} \{g(y)\}^k dS(y|X_i)$ in (9) as

$$\sum_{l=1}^{m_v - 1} \{g(v_l)\}^k \left\{\tilde{\lambda}(v_{l+1}|X_i) - \tilde{\lambda}(v_l|X_i)\right\} I(u_{j-1} \le v_{l-1} < v_l \le u_j).$$

# D    PROOFS OF THEORETICAL RESULTS

*Proof of Proposition 1.*

$$
\begin{aligned}
E\{L_{\text{CUT}}(\mathcal{O}_i, f(X_i))\} &= E\left[\frac{1}{2}\tilde{Y}_2(\mathcal{O}_i) - \tilde{Y}_1(\mathcal{O}_i)f(X_i) + \frac{1}{2}\{f(X_i)\}^2\right] \\
&= \frac{1}{2}E\left\{\tilde{Y}_2(\mathcal{O}_i)\right\} - E\left\{\tilde{Y}_1(\mathcal{O}_i)f(X_i)\right\} + \frac{1}{2}E\left[\{f(X_i)\}^2\right] \\
&= \frac{1}{2}E\left(\sum_{j=1}^{m+1}\Delta_{i,j}E\left[\{g(Y_i)\}^2|\Delta_{i,j}=1,X_i\right]\right) \\
&\quad - E\left[f(X_i)\sum_{j=1}^{m+1}\Delta_{i,j}E\left\{g(Y_i)|\Delta_{i,j}=1,X_i\right\}\right] + \frac{1}{2}E\left[\{f(X_i)\}^2\right] \\
&= \frac{1}{2}E\left(E\left[\{g(Y_i)\}^2|X_i\right]\right) - E\left[f(X_i)E\left\{g(Y_i)|X_i\right\}\right] + E\left[\frac{1}{2}\{f(X_i)\}^2\right] \\
&= E\left(\frac{1}{2}E\left[\{g(Y_i)\}^2|X_i\right]\right) - E\left[E\left\{f(X_i)g(Y_i)|X_i\right\}\right] + E\left[\frac{1}{2}\{f(X_i)\}^2\right] \\
&= E\left[\frac{1}{2}\{g(Y_i)\}^2\right] - E\left\{f(X_i)g(Y_i)\right\} + E\left[\frac{1}{2}\{f(X_i)\}^2\right] \\
&= E\{L(Y_i, f(X_i))\},
\end{aligned}
$$

where the first step uses (10), the third step is due to (8), the fourth and six steps come from the law of total expectation, the fifth step is from the the property of conditional expectation, and the last step uses (7). □

To prove Proposition 2 - 6, Corollary 1, and Theorems 1 - 5, we adapt the techniques of Bühlmann & Yu (2003) with modifications tailored to our specific setup.

*Proof of Proposition 2.* For $\vec{f}^{(0)}$ in Line 1 of Algorithm 1, we choose $\Psi$ such that

$$\vec{f}^{(0)} = \Psi\vec{Y}_1. \tag{D.1}$$

By (13), we obtain that for $t = 1, 2, \ldots$,

$$
\begin{aligned}
\vec{u}^{(t-1)} &= \vec{Y}_1 - \vec{f}^{(t-1)} \\
&= \vec{Y}_1 - \left(\vec{f}^{(t)} - \vec{h}^{(t)}\right) \\
&= \vec{Y}_1 - \vec{f}^{(t)} + \Psi\vec{u}^{(t-1)} \\
&= \vec{u}^{(t)} + \Psi\vec{u}^{(t-1)},
\end{aligned}
$$

where the second step is due to Line 5 of Algorithm 1, the third step comes from (14), and the last step is due to (13). Therefore,

$$\vec{u}^{(t)} = (I - \Psi)\vec{u}^{(t-1)}. \tag{D.2}$$

Recursively applying (D.2), we have that for $t = 1, 2\ldots$,

$$
\begin{aligned}
\vec{u}^{(t-1)} &= (I - \Psi)^{t-1}\vec{u}^{(0)} \\
&= (I - \Psi)^{t-1}\left(\vec{Y}_1 - \vec{f}^{(0)}\right) \\
&= (I - \Psi)^{t-1}\left(\vec{Y}_1 - \Psi\vec{Y}_1\right) \\
&= (I - \Psi)^t\vec{Y}_1, \tag{D.3}
\end{aligned}
$$

where the second step uses (13) and the third step is due to (D.1).

Recursively applying Line 5 of Algorithm 1, we have that for $t = 1, 2 \ldots,$

$$
\begin{aligned}
\vec{f}^{(t)} &= \vec{f}^{(0)} + \sum_{j=1}^{t} \vec{h}^{(j)} \\
&= \Psi \vec{Y}_1 + \sum_{j=1}^{t} \Psi (I - \Psi)^j \vec{Y}_1 \\
&= \sum_{j=0}^{t} \Psi (I - \Psi)^j \vec{Y}_1 \\
&= \left\{ I - (I - \Psi)^{t+1} \right\} \vec{Y}_1,
\end{aligned}
$$

where the second step uses (14), (D.1), and (D.3); and the last step comes from the fact that for a symmetric matrix $A$ with $(I - A)$ being invertible, $\sum_{j=0}^{t} A^j = (I - A)^{-1} \left( I - A^{t+1} \right)$, which may be derived using the same reasoning for geometric series.

Therefore, by (16), we may set that $B^{(t)} = I - (I - \Psi)^{t+1}$. $\qquad \square$

*Proof of Proposition 3.* We examine the MSE in (17):

$$
\begin{aligned}
\text{MSE}(t, \Psi; \phi) &= n^{-1} \sum_{i=1}^{n} E \left[ \left\{ f^{(t)}(X_i) - \phi(X_i) \right\}^2 \right] \\
&= n^{-1} \sum_{i=1}^{n} \left( \text{var} \left\{ f^{(t)}(X_i) - \phi(X_i) \right\} + \left[ E \left\{ f^{(t)}(X_i) - \phi(X_i) \right\} \right]^2 \right) \\
&= n^{-1} \sum_{i=1}^{n} \left[ \text{var} \left( f^{(t)}(X_i) \right) + \left\{ E \left( f^{(t)}(X_i) \right) - \phi(X_i) \right\}^2 \right] \\
&= n^{-1} \sum_{i=1}^{n} \text{var} \left( f^{(t)}(X_i) \right) + n^{-1} \sum_{i=1}^{n} \left\{ E \left( f^{(t)}(X_i) \right) - \phi(X_i) \right\}^2,
\end{aligned}
$$

where the second step comes from the property that $E(U^2) = \text{var}(U) + \{E(U)\}^2$ for any random variable $U$, and the third step dues to the fact $\phi(X_i)$ is taken as a constant. $\qquad \square$

To prove Proposition 4, we use the following basic properties of matrices.

**Lemma 1.** *Let $A$ and $B$ be two symmetric matrices of the same dimension. Let $I$ be the identity matrix of the same dimension as $A$. Then the following results hold:*

- *(a). $A + B$ and $A - B$ are symmetric matrices;*

- *(b). $A^k$ is symmetric for any integer $k$;*

- *(c). If $A$ has eigenvalues $\{\lambda_1, \ldots, \lambda_n\}$ and corresponding normalized eigenvectors $\{Q_1, \ldots, Q_n\}$. Then*

  - *(i). for any positive integer $k$, the eigenvalues of $A^k$ are $\{\lambda_1^k, \ldots, \lambda_n^k\}$ with $\{Q_1, \ldots, Q_n\}$ being the corresponding eigenvectors;*
  - *(ii). the eigenvalues of $A + I$ are $\{\lambda_1 + 1, \ldots, \lambda_n + 1\}$ with $\{Q_1, \ldots, Q_n\}$ being the corresponding eigenvectors;*
  - *(iii). the eigenvalues of $-A$ are $\{-\lambda_1, \ldots, -\lambda_n\}$ with $\{Q_1, \ldots, Q_n\}$ being the corresponding eigenvectors.*

*Proof of Proposition 4.* Assume that $\Psi$ is symmetric with eigenvalues $\{\lambda_1, \ldots, \lambda_n\}$ and corresponding normalized eigenvectors $\{Q_1, \ldots, Q_n\}$. By Proposition 2 together with Lemma 1, we have that $B^{(t)}$ is also symmetric, and its eigenvalues are $\left\{ 1 - (1 - \lambda_1)^{t+1}, \ldots, 1 - (1 - \lambda_n)^{t+1} \right\}$

with corresponding eigenvectors $\{Q_1, \ldots, Q_n\}$. Consequently, we can decompose $B^{(t)}$ using orthonormal diagonalization as

$$B^{(t)} = Q\Lambda^{(t)}Q^{-1}, \tag{D.4}$$

where $\Lambda^{(t)} \triangleq \operatorname{diag}\{1 - (1 - \lambda_k)^{t+1} : k = 1, \ldots, n\}$ and the matrix $Q \triangleq (Q_1, \ldots, Q_n)$ is orthonormal, satisfying $QQ^\top = Q^\top Q = I$ and $Q^{-1} = Q^\top$.

Next, we examine the variance in (18):

$$
\begin{aligned}
\operatorname{var}(t, \Psi) &= n^{-1} \sum_{i=1}^{n} \operatorname{var}\left\{ f^{(t)}(X_i) \right\} \\
&= n^{-1}\operatorname{tr}\left\{ \operatorname{cov}\left( \vec{f}^{(t)} \right) \right\} \\
&= n^{-1}\operatorname{tr}\left\{ \operatorname{cov}\left( B^{(t)}\vec{Y}_1 \right) \right\} \\
&= n^{-1}\operatorname{tr}\left\{ B^{(t)}\operatorname{cov}\left( \vec{Y}_1 \right) \left( B^{(t)} \right)^\top \right\} \\
&= n^{-1}\operatorname{tr}\left\{ Q\Lambda^{(t)}Q^{-1}\hat{\sigma}^2 I \left( Q\Lambda^{(t)}Q^{-1} \right)^\top \right\} \\
&= \hat{\sigma}^2 n^{-1}\operatorname{tr}\left( Q \operatorname{diag}\left[ \left\{ 1 - (1 - \lambda_k)^{t+1} \right\}^2 : k = 1, \ldots, n \right] Q^\top \right) \\
&= \hat{\sigma}^2 n^{-1} \sum_{k=1}^{n} \left\{ 1 - (1 - \lambda_k)^{t+1} \right\}^2,
\end{aligned}
\tag{D.5}
$$

where the second step follows from the definition of the trace of the covariance matrix, the third step is from (16), the fourth step applies the property of scaling the covariance matrix when multiplied by a constant matrix, the fifth step uses (D.4) and the definition of $\hat{\sigma}^2$, the sixth step is derived from the properties of the trace and the fact that $Q^\top Q = I$, and the final step follows from the matrix product with a diagonal matrix and $Q^\top Q = I$.

Finally, we examine the squared bias, given in (18):

$$
\begin{aligned}
\operatorname{bias}^2(t, \Psi; \phi) &= n^{-1} \sum_{i=1}^{n} \left[ E\left\{ f^{(t)}(X_i) \right\} - \phi(X_i) \right]^2 \\
&= n^{-1} \left\{ E\left( B^{(t)}\vec{Y}_1 \right) - \vec{\phi} \right\}^\top \left\{ E\left( B^{(t)}\vec{Y}_1 \right) - \vec{\phi} \right\} \\
&= n^{-1} \left\{ \left( B^{(t)} - I \right) \vec{\phi} \right\}^\top \left\{ \left( B^{(t)} - I \right) - \vec{\phi} \right\} \\
&= n^{-1} \left[ \left\{ Q\left( \Lambda^{(t)} - I \right) Q^{-1} \right\} \vec{\phi} \right]^\top \left[ \left\{ Q\left( \Lambda^{(t)} - I \right) Q^{-1} \right\} \vec{\phi} \right] \\
&= n^{-1} \vec{\phi}^\top Q \left( \Lambda^{(t)} - I \right)^\top \left( \Lambda^{(t)} - I \right) Q^\top \vec{\phi} \\
&= n^{-1} \vec{\phi}^\top Q \operatorname{diag}\left\{ (1 - \lambda_l)^{2t+2} : l = 1, \ldots, n \right\} Q^\top \vec{\phi} \\
&= n^{-1} \sum_{l=1}^{n} \mu_l^2 (1 - \lambda_l)^{2t+2},
\end{aligned}
\tag{D.6}
$$

where the second step is due to (16), the third step is due to $E\left\{ \hat{Y}_1(\mathcal{O}_i) \right\} = E(Y_i) = \phi(X_i)$, the fourth step uses (D.4), the fifth step comes from $Q^\top Q = I$ and $Q^{-1} = Q^\top$, and the last step is due to definition of $\mu$, given before Proposition 4. □

*Proof of Corollary 1.* This corollary follows directly from using the properties of diagonal matrices that have entries either 0 or 1. □

*Proof of Proposition 5.* By condition (C2), $0 \leq (1 - \lambda_l) < 1$, and thus, $\operatorname{bias}^2(t, \Psi; \phi)$ in (D.6) decays exponentially with increasing $t$ and $\operatorname{var}(t, \Psi)$ in (D.5) exhibits an exponentially small increase

as $t$ increases. Further, by (D.5), we have that

$$\lim_{t \to \infty} \text{var}(t, \Psi) = \lim_{t \to \infty} \hat{\sigma}^2 n^{-1} \sum_{l=1}^{n} \left\{ 1 - (1 - \lambda_l)^{t+1} \right\}^2$$

$$= \hat{\sigma}^2 n^{-1} \sum_{l=1}^{n} \left\{ 1 - \lim_{t \to \infty} (1 - \lambda_l)^{t+1} \right\}^2$$

$$= \hat{\sigma}^2 n^{-1} \sum_{l=1}^{n} 1$$

$$= \hat{\sigma}^2,$$

and by (D.6), we obtain that

$$\lim_{t \to \infty} \text{bias}^2(t, \Psi; \phi) = \lim_{t \to \infty} n^{-1} \sum_{l=1}^{n} \mu_l^2 (1 - \lambda_l)^{2t+2}$$

$$= n^{-1} \sum_{l=1}^{n} \mu_l^2 \lim_{t \to \infty} (1 - \lambda_l)^{2t+2}$$

$$= n^{-1} \sum_{l=1}^{n} \mu_l^2 0$$

$$= 0.$$

Therefore, by Proposition 3,

$$\lim_{t \to \infty} \text{MSE}(t, \Psi; \phi) = \lim_{t \to \infty} \text{var}(t, \Psi) + \lim_{t \to \infty} \text{bias}^2(t, \Psi; \phi)$$

$$= \hat{\sigma}^2.$$

$\square$

*Proof of Proposition 6.* By propositions 3 and 4, we obtain that

$$\text{MSE}(t, \Psi; \phi) = \hat{\sigma}^2 n^{-1} \sum_{l=1}^{n} \left\{ 1 - (1 - \lambda_l)^{t+1} \right\}^2 + n^{-1} \sum_{l=1}^{n} \mu_l^2 (1 - \lambda_l)^{2t+2}.$$

Considering this as a function of $t$ only, with other quantities treated as fixed, we consider the function:

$$\psi(u) \triangleq \hat{\sigma}^2 n^{-1} \sum_{l=1}^{n} \left\{ 1 - (1 - \lambda_l)^{u+1} \right\}^2 + n^{-1} \sum_{l=1}^{n} \mu_l^2 (1 - \lambda_l)^{2u+2},$$

which equals

$$\psi(u) = \hat{\sigma}^2 n^{-1} \sum_{l=1}^{n} \left\{ 1 - 2(1 - \lambda_l)^{u+1} + (1 - \lambda_l)^{2u+2} \right\} + n^{-1} \sum_{l=1}^{n} \mu_l^2 (1 - \lambda_l)^{2u+2}$$

$$= n^{-1} \sum_{l=1}^{n} \left\{ \left( \hat{\sigma}^2 + \mu_l^2 \right) (1 - \lambda_l)^{u+1} - 2\hat{\sigma}^2 \right\} (1 - \lambda_l)^{u+1} + \hat{\sigma}^2$$

$$= n^{-1} \sum_{k:\lambda_k < 1} \left\{ \left( \hat{\sigma}^2 + \mu_k^2 \right) (1 - \lambda_k)^{u+1} - 2\hat{\sigma}^2 \right\} (1 - \lambda_k)^{u+1} + \hat{\sigma}^2. \quad \text{(D.7)}$$

By condition (C3), there exists at least one $k$ such that $\lambda_k < 1$. Considering all those $k$ such that $\lambda_k < 1$, we let $k_1, \ldots, k_{n_0}$ denote them, where $n_0 \le n$. For $j = 1, \ldots, n_0$,

$$\lim_{u \to \infty} \left\{ \left( \hat{\sigma}^2 + \mu_{k_j}^2 \right) (1 - \lambda_{k_j})^{u+1} - 2\hat{\sigma}^2 \right\} = -2\hat{\sigma}^2,$$

leading to

$$\lim_{u \to \infty} \left\{ \left( \hat{\sigma}^2 + \mu_{k_j}^2 \right) (1 - \lambda_{k_j})^{u+1} - 2\hat{\sigma}^2 \right\} < -\hat{\sigma}^2.$$

Therefore, for $j = 1, \ldots, n_0$, there exists $u_j$ such that

$$\left(\hat{\sigma}^2 + \mu_{k_j}^2\right)\left(1 - \lambda_{k_j}\right)^{u_j+1} - 2\hat{\sigma}^2 < -\hat{\sigma}^2. \tag{D.8}$$

Letting $t_0 = \max(u_1, \ldots, u_{n_0})$, (D.8) yields that for $j = 1, \ldots, n_0$,

$$\left(\hat{\sigma}^2 + \mu_{k_j}^2\right)\left(1 - \lambda_{k_j}\right)^{t_0+1} - 2\hat{\sigma}^2 < -\hat{\sigma}^2.$$

Therefore, (D.7) leads to

$$\psi(u) < -n^{-1}\sum_{j=1}^{n_0} \hat{\sigma}^2(1 - \lambda_{k_j})^{t_0+1} + \hat{\sigma}^2 < \hat{\sigma}^2,$$

and the conclusion follows. $\qquad\qquad\square$

*Proof of Theorem 1.* We calculate the derivative of $\psi(u)$ in (D.7):

$$\psi'(u) = 2n^{-1}\sum_{k:\lambda_k<1}^{n} \left\{\left(\hat{\sigma}^2 + \mu_k^2\right)\left(1 - \lambda_k\right)^{u+1} - \hat{\sigma}^2\right\}\left(1 - \lambda_k\right)^{u+1}\log(1 - \lambda_k).$$

Now consider those $k$ with $\lambda_k < 1$, condition (C4) leads to $\left(\hat{\sigma}^2 + \mu_k^2\right)\left(1 - \lambda_k\right)^{m_0} > \hat{\sigma}^2$. Since for any $u \in (0, m_0 - 1)$, $(1 - \lambda_k)^{m_0} < (1 - \lambda_k)^{u+1}$, which yields that $\left(\hat{\sigma}^2 + \mu_k^2\right)(1 - \lambda_k)^{u+1} > \hat{\sigma}^2$ for any $u \in (0, m_0 - 1)$. Therefore, $\psi'(u) < 0$ for all $u \in (0, m_0 - 1)$. $\psi(u)$ is decreasing over $(0, m_0 - 1)$. By the continuity of $\psi(u)$ over $[0, m_0 - 1]$, we have that $\psi(0) > \psi(1) > \ldots > \psi(m_0 - 1)$, suggesting that the first $\lfloor m_0 - 1 \rfloor$ iterations of the $L_2$Boost-CUT algorithm improves the MSE over the unboosted base learner algorithm (i.e., corresponding to $\psi(0)$). $\qquad\square$

*Proof of Theorem 2.* For a vector $u$, we use $(u)_i$, $\{u\}_i$, or $[u]_i$ to denote its $i$th element. For $i = 1, \ldots, n$, let $b^{(t)}(X_i) \triangleq E\left\{f^{(t)}(X_i)\right\} - \phi(X_i)$ denote the bias term for subject $i$. Let $\vec{\epsilon} = (\hat{\epsilon}_1, \ldots, \hat{\epsilon}_n)$.

We examine the summands of the left-hand side of (19):

$$E\left[\left\{f^{(t)}(X_i) - \phi(X_i)\right\}^q\right]$$
$$= E\left(\left[f^{(t)}(X_i) - E\left\{f^{(t)}(X_i)\right\} + E\left\{f^{(t)}(X_i)\right\} - \phi(X_i)\right]^q\right)$$
$$= E\left(\sum_{l=0}^{q}\binom{q}{l}\left[E\left\{f^{(t)}(X_i)\right\} - \phi(X_i)\right]^l\left[f^{(t)}(X_i) - E\left\{f^{(t)}(X_i)\right\}\right]^{q-l}\right)$$
$$= E\left(\sum_{l=0}^{q}\binom{q}{l}\left\{b^{(t)}(X_i)\right\}^l\left[\left(B^{(t)}\hat{Y}_1\right)_i - E\left\{\left(B^{(t)}\hat{Y}_1\right)_i\right\}\right]^{q-l}\right)$$
$$= E\left[\sum_{l=0}^{q}\binom{q}{l}\left\{b^{(t)}(X_i)\right\}^q\left\{\left(B^{(t)}\vec{\epsilon}\right)_i\right\}^{q-l}\right]$$
$$= \sum_{l=0}^{q}\binom{q}{l}\left\{b^{(t)}(X_i)\right\}^l E\left[\left\{\left(B^{(t)}\vec{\epsilon}\right)_i\right\}^{q-l}\right]$$
$$= \sum_{l=0}^{q}\binom{q}{l}\left\{b^{(t)}(X_i)\right\}^l E\left\{\left(\left[\left\{I - (I - \Psi)^{t+1}\right\}\vec{\epsilon}\right]_i\right)^{q-l}\right\}$$
$$= \sum_{l=0}^{q}\binom{q}{l}\left\{b^{(t)}(X_i)\right\}^l E\left(\left[\left\{\vec{\epsilon} - (I - \Psi)^{t+1}\vec{\epsilon}\right\}_i\right]^{q-l}\right), \tag{D.9}$$

where the third step is due to (16); the fourth step is due to the definition of $\hat{\epsilon}_i$; the fifth step is derived under assumption that $X_i$ is treated as a constant; and the sixth step is due to Proposition 2.

Then, we examine $b^{(t)}(X_i)$:

$$
\begin{aligned}
b^{(t)}(X_i) &= E\left\{ f^{(t)}(X_i) \right\} - \phi(X_i) \\
&= \left\{ E\left( B^{(t)}\vec{Y_1} \right) - \vec{\phi} \right\}_i \\
&= \left\{ \left( B^{(t)} - I \right) \vec{\phi} \right\}_i \\
&= \left[ \left\{ Q\left( \Lambda^{(t)} - I \right) Q^{-1} \right\} \vec{\phi} \right]_i \\
&= \left[ Q \operatorname{diag}\left\{ -(\lambda_l - 1)^{t+1} : l = 1, \ldots, n \right\} Q^\top \vec{\phi} \right]_i \\
&= -\sum_{k=1}^{n} Q_{ik}(\lambda_k - 1)^{t+1}\mu_k \\
&= O\left( \exp(-C_b t) \right) \text{ as } t \to \infty,
\end{aligned}
\tag{D.10}
$$

for some positive constant $C_b$, where the second step is due to (16), the third step is due to $E\left\{ \hat{Y_1}(\mathcal{O}_i) \right\} = E(Y_i) = \phi(X_i)$, the fourth step is from (D.4), the fifth step comes from the definition of $\Lambda^{(t)}$, and the six step comes from the definition of $\mu = Q^\top \vec{u}$, with $Q_{ik}$ representing the $(i, k)$th element of $Q$.

Next, we examine $E\left( \left[ \left\{ \vec{\epsilon} - (I - \Psi)^{t+1}\vec{\epsilon} \right\}_i \right]^{q-l} \right)$:

$$
\begin{aligned}
&E\left( \left[ \left\{ \vec{\epsilon} - (I - \Psi)^{t+1}\vec{\epsilon} \right\}_i \right]^{q-l} \right) \\
&= E\left( \sum_{k=0}^{q-l} \binom{q-l}{k} \hat{\epsilon}_i^k \left\{ (I-\Psi)^{t+1}\vec{\epsilon} \right\}_i^{q-l-k} \right) \\
&= E\left( \sum_{k=0}^{q-l} \binom{q-l}{k} \hat{\epsilon}_i^k \left[ \left\{ Q\operatorname{diag}(1-\lambda_j : j=1,\ldots,n)Q^{-1} \right\}^{t+1} \vec{\epsilon} \right]_i^{q-l-k} \right) \\
&= E\left( \sum_{k=0}^{q-l} \binom{q-l}{k} \hat{\epsilon}_i^k \left[ Q\operatorname{diag}\left\{ (1-\lambda_j)^{t+1} : j=1,\ldots,n \right\} Q^{-1}\vec{\epsilon} \right]_i^{q-l-k} \right) \\
&= E\left[ \sum_{k=0}^{q-l} \binom{q-l}{k} \hat{\epsilon}_i^k \left\{ \sum_{j=1}^{n} Q_{ij}(1-\lambda_j)^{t+1} \left( \sum_{u=1}^{n} Q_{ju}^{-1}\hat{\epsilon}_u \right) \right\}^{q-l-k} \right] \\
&= E(\hat{\epsilon}_i^{q-l}) + O\left( \exp(-C_q t) \right) \text{ as } t \to \infty
\end{aligned}
\tag{D.11}
$$

for some positive constant $C_q$.

Combining (D.9) with (D.10) and (D.11) yields

$$
\begin{aligned}
&n^{-1} \sum_{i=1}^{n} E\left[ \left\{ f^{(t)}(X_i) - \phi(X_i) \right\}^q \right] \\
&= n^{-1} \sum_{i=1}^{n} \sum_{l=0}^{q} \binom{q}{l} \left\{ O\left( \exp(-C_b t) \right) \right\}^l \left\{ E(\hat{\epsilon}_i^{q-l}) + O\left( \exp(-C_q t) \right) \right\} \\
&= E(\hat{\epsilon}_i^q) + O\left( \exp(-C t) \right)
\end{aligned}
$$

for some positive constant $C$. $\qquad\square$

*Proof of Theorem 3.* Let $\Psi$ denote the smoother matrix for the smoothing spline of degree $r$ and degrees of freedom $df$ (equivalently expressed in terms of tuning parameter $\lambda$). Given the tuning parameter $\lambda$, the eigenvalues of $\Psi$ are arranged in decreasing order and are written as:

$$
\lambda_1 = \ldots = \lambda_r = 1, \quad \lambda_l = \frac{q_{l,n}}{\lambda + q_{l,n}} \text{ for } l = r+1, \ldots, n,
\tag{D.12}
$$

where $q_{l,n}$ depends on $\Omega$ defined in Appendix B (Utreras, 1983; Bühlmann & Yu, 2003; Hastie et al., 2009).

By Assumption (A) of Bühlmann & Yu (2003), Utreras (1988) showed that for large $n$, there exists a finite positive constant $a_0$ such that

$$q_{l,n} \approx a_0 l^{-2r}. \tag{D.13}$$

For $f \in \mathcal{W}_2^{(v)}[a, b]$, there exists a finite positive constant $M$ such that

$$n^{-1} \sum_{l=r+1}^{n} \mu_l^2 l^{2v} \le M. \tag{D.14}$$

Let $\tilde{\lambda} = \lambda/a_0$. Then by (D.13), the $\lambda_l$ for $l = r+1, \ldots, n$ in (D.12) are

$$\lambda_l \approx \frac{l^{-2r}}{\tilde{\lambda} + l^{-2r}} = \frac{1}{\tilde{\lambda}l^{2r} + 1}. \tag{D.15}$$

Then (D.6) can be bounded by

$$
\begin{aligned}
\text{bias}^2(t, \Psi; \phi) &= n^{-1} \sum_{l=r+1}^{n} \mu_l^2 (1 - \lambda_l)^{2t+2} \\
&\approx n^{-1} \sum_{l=r+1}^{n} \mu_l^2 l^{2v} \left(1 - \frac{1}{\tilde{\lambda}l^{2r} + 1}\right)^{2t+2} l^{-2v} \\
&\le \left\{ \max_{l=r+1,\ldots,n} \left(1 - \frac{1}{\tilde{\lambda}l^{2r} + 1}\right)^{2t+2} l^{-2v} \right\} n^{-1} \sum_{l=r+1}^{n} \mu_l^2 l^{2v} \\
&\le M \left\{ \max_{l=r+1,\ldots,n} \left(1 - \frac{1}{\tilde{\lambda}l^{2r} + 1}\right)^{2t+2} l^{-2v} \right\} \\
&\triangleq M \max_{l=r+1,\ldots,n} \exp\{\eta(l)\},
\end{aligned}
\tag{D.16}
$$

with

$$\eta(l) = (2t + 2) \log \left(1 - \frac{1}{\tilde{\lambda}l^{2r} + 1}\right) - 2v \log(l), \tag{D.17}$$

where the second step uses (D.15), and the fourth step uses (D.14). Taking the derivative of (D.17) yields

$$
\begin{aligned}
\eta'(l) &= \frac{2r(2t + 2)}{l(\tilde{\lambda}l^{2r} + 1)} - \frac{2v}{l} \\
&= \frac{2r}{l(\tilde{\lambda}l^{2r} + 1)} \left\{ 2t + 2 - \frac{v(\tilde{\lambda}l^{2r} + 1)}{r} \right\}.
\end{aligned}
$$

Now, consider any positive integer $n_1$ with $r < n_1 \le n$, and

$$t \ge \{v(\tilde{\lambda}n_1^{2r} + 1)\}/(2r) - 1. \tag{D.18}$$

Then $\eta'(l) \ge 0$ for any $0 < l \le n_1$, therefore, $\eta(l)$ is increasing and so is $\exp\{\eta(l)\}$ for $0 < l \le n_1$. Therefore, for any $r < l \le n_1$, we have that

$$
\begin{aligned}
\exp\{\eta(l)\} &\le \exp\{\eta(n_1)\} \\
&= \left(1 - \frac{1}{\tilde{\lambda}n_1^{2r} + 1}\right)^{2t+2} n_1^{-2v} \\
&\le \left(1 - \frac{1}{\tilde{\lambda}n^{2r} + 1}\right)^{2t+2} n_1^{-2v}.
\end{aligned}
\tag{D.19}
$$

Applying (D.19) to (D.16) gives that for $n_1$ and $t$ in (D.18), and $t \geq \{v(\tilde{\lambda}n_1^{2r} + 1)\}/(2r) - 1$, we have that

$$\text{bias}^2(t, \Psi; \phi) \leq M \left(1 - \frac{1}{\tilde{\lambda}n^{2r} + 1}\right)^{2t+2} n_1^{-2v}$$

$$\leq M n_1^{-2v} \text{ as } n_1 \to \infty, \tag{D.20}$$

and hence, $\text{bias}^2(t, \Psi; \phi)$ is of order $O\left(n_1^{-2v}\right)$ as $n_1 \to \infty$.

Now we examine (D.5). For any $n_1$ in (D.18), by (D.12),

$$\text{var}(t, \Psi) = \frac{\hat{\sigma}^2}{n} \left[r + \sum_{l=r+1}^{n} \left\{1 - (1 - \lambda_l)^{t+1}\right\}^2\right]$$

$$\leq \frac{\hat{\sigma}^2 n_1}{n} + \frac{\hat{\sigma}^2}{n} \sum_{l=n_1+1}^{n} \left\{1 - (1 - \lambda_l)^{t+1}\right\}^2$$

$$= O\left(\frac{n_1}{n}\right) + \frac{\hat{\sigma}^2}{n} \sum_{l=n_1+1}^{n} \left\{1 - (1 - \lambda_l)^{t+1}\right\}^2. \tag{D.21}$$

By Bernoulli's inequality that $(1 - a)^b \geq 1 - ab$ for $a \leq 1$ and $b \geq 0$, we obtain that

$$1 - (1 - \lambda_l)^{t+1} \leq 1 - \{1 - \lambda_l(t+1)\} = \lambda_l(t+1).$$

Therefore, for $t$ in (D.18), by (D.15), we obtain that

$$\frac{\hat{\sigma}^2}{n} \sum_{l=n_1+1}^{n} \left\{1 - (1 - \lambda_l)^{t+1}\right\}^2$$

$$\leq \frac{\hat{\sigma}^2}{n} \sum_{l=n_1+1}^{n} \lambda_l^2(t+1)^2$$

$$\approx \frac{\hat{\sigma}^2(t+1)^2}{n} \sum_{l=n_1+1}^{n} \frac{1}{\left(\tilde{\lambda}l^{2r} + 1\right)^2}$$

$$\leq \frac{\hat{\sigma}^2(t+1)^2}{n} \sum_{l=n_1+1}^{n} \frac{1}{\left(\tilde{\lambda}l^{2r}\right)^2}$$

$$\leq \frac{\hat{\sigma}^2(t+1)^2}{n} \int_{n_1}^{\infty} \frac{1}{\left(\tilde{\lambda}u^{2r}\right)^2} du$$

$$= \frac{\hat{\sigma}^2(t+1)^2}{\tilde{\lambda}^2(4r-1)} \left(\frac{n_1^{1-4r}}{n}\right)$$

$$\leq O\left(\frac{n_1}{n}\right) \text{ as } n_1 \to \infty. \tag{D.22}$$

Applying (D.20), (D.21), and (D.22) to Proposition 3, we obtain that

$$\text{MSE}(t, \Psi; \phi) \leq O\left(\frac{n_1}{n}\right) + O\left(n_1^{-2v}\right) \text{ as } n_1 \to \infty.$$

Treating the order as a function of $n_1$, it is minimized as $O\left(n^{-2v/(2v+1)}\right)$ by taking $n_1 = O\left(n^{1/(2v+1)}\right)$. Therefore, for this $n_1$, $t$ in (D.18) can be taken as $t_n \triangleq O\left(n^{2r/(2v+1)}\right)$. $\qquad\square$

*Proof of Theorem 5.* By Theorem 2.3 and the discussion on Page 102 of Devroye et al. (1996), we have that

$$n^{-1} \sum_{i=1}^{n} \Pr\left(f_s^{(t_n)} \neq Y_i\right) - \text{BR} \leq 2\sqrt{\text{MSE}(t, \Psi; f_s)}$$

$$= O\left(n^{-v/(2v+1)}\right) \text{ as } n \to \infty,$$

where the last step is due to Theorem 4. $\qquad\square$

# E DISCUSSIONS AND EXTENSIONS

## E.1 LEARNING RATE FOR $L_2$BOOST AND OUR PROPOSED ALGORITHMS

In traditional boosting methods, a learning rate, denoted as $\hat{\alpha}^{(t)}$, is introduced to control the contribution of $h^{(t)}$ at each iteration $t$ for $t = 1, 2, \ldots$, scaling how much it corrects the prediction error of $f^{(t)}$:

$$f^{(t)} = f^{(t-1)} + \hat{\alpha}^{(t)}\hat{h}^{(t)},$$

where

$$\hat{\alpha}^{(t)} = \underset{\alpha^{(t)} \in \mathbb{R}}{\arg\min} \left\{ n^{-1} \sum_{i=1}^{n} L\left(Y_i, f^{(t-1)}(X_i) + \alpha^{(t)}\hat{h}^{(t)}(X_i)\right) \right\}.$$

However, in our algorithms, which are based on the $L_2$ loss function, the learning rate $\hat{\alpha}^{(t)}$ is inherently incorporated within the optimization process for $\hat{h}^{(t)}$. Specifically, when using the $L_2$ loss, the minimization problem for $\hat{\alpha}^{(t)}$ simplifies to

$$\hat{\alpha}^{(t)} = \underset{\alpha^{(t)} \in \mathbb{R}}{\arg\min} \left[ n^{-1} \sum_{i=1}^{n} \left\{ Y_i - f^{(t-1)}(X_i) - \alpha^{(t)}\hat{h}^{(t)}(X_i) \right\}^2 \right],$$

which is integrated naturally into the computation of $\hat{h}^{(t)}$ because of (11):

$$\hat{h}^{(t)} = \underset{h^{(t)}}{\arg\min} \left[ n^{-1} \sum_{i=1}^{n} \left\{ Y_i - f^{(t-1)}(X_i) - h^{(t)}(X_i) \right\}^2 \right].$$

As a result, Algorithm 1 does not require an explicit learning rate parameter, as $\hat{\alpha}^{(t)}$ is effectively determined as part of the optimization of $\hat{h}^{(t)}$.

## E.2 COMPUTATIONAL COMPLEXITY

Our proposed $L_2$Boost-CUT method in Algorithm 1 basically comprises two components: ICRF and boosting. The computational complexity of ICRF is $O(n^\gamma)$, with $1 < \gamma \leq 2$ (Cho et al., 2022). For smoothing splines, when implemented efficiently, the complexity can be $O(n)$ (Hastie et al., 2009, Chapter 5). With $\tilde{t}$ boosting iterations and smoothing splines as base learners, the total computational complexity is $O(\tilde{t}n)$. Therefore, the overall computational complexity of the $L_2$Boost-CUT method is $O(n^\gamma + \tilde{t}n)$.

## E.3 POSSIBLE EXTENSIONS

The $L_2$Boost-CUT framework can be extended to $L_q$ loss functions for handling interval-censored data, where $q > 2$ is an integer, and the $L_q$ loss function is given by

$$L\left(g(Y_i), f(X_i)\right) \propto \{g(Y_i) - f(X_i)\}^q$$

$$= \sum_{k=0}^{q} \binom{q}{k} \{g(Y_i)\}^k \{-f(X_i)\}^{q-k},$$

with $\{g(Y_i)\}^k$ replaced by its transformed form (8), together with (9). Here, $k$ is extended to take any value in $\{1, \ldots, q\}$.

As shown in (11), the linear derivative of the $L_2$ loss with respect to its first argument suggests closely related increment terms in both $L_2$Boost-CUT and $L_2$Boost-IMP, thus often leading to similar results. However, this connection does not hold for the loss function $L_q$ when $q \geq 3$. Consequently, the $L_q$Boost-CUT and $L_q$Boost-IMP methods likely yield more different results, where the $L_q$Boost-IMP method is obtained by replacing the $L_2$ loss in the $L_2$Boost-IMP method with the $L_q$ loss.

While extending the $L_2$Boost-CUT method to accommodate the $L_q$ loss for $q \geq 3$ is straightforward, adapting it to any general loss function for constructing an adjusted loss function like $L_{\mathrm{CUT}}$ in

(10) to ensure Proposition 1 presents significant challenges. In contrast, generalizing the $L_2$Boost-IMP method to any loss function is straightforward by using imputed values determined by the transformed response in (8).

For example, considering widely used loss functions, such as exponential loss function $L(u, v) = \exp(-uv)$ (Schapire & Singer, 1998) and the binomial deviance loss $L(u, v) = \log\{1 + \exp(-2uv)\}$ (Friedman et al., 2000), one may apply the censoring-unbiased transformation (8) to these loss functions and adapt the proposed $L_2$Boost-IMP method to enable boosting algorithms like AdaBoost (Freund & Schapire, 1996) and LogitBoost (Friedman et al., 2000) to handle interval-censored data. For XGBoost (Chen & Guestrin, 2016), one may replace $l$ in (2) of Chen & Guestrin (2016) with the transformed unbiased loss function (10). This extension would allow XGBoost to handle interval-censored data.

While $L_2$Boost-CUT can be extended to boosting frameworks with $L_q$ losses ($q \geq 3$) and $L_2$Boost-IMP can be extended to accommodate any loss function procedurally, establishing theoretical properties for these extensions is nontrival. Unlike the $L_2$Boost-CUT method, optimal learning rates would need to be estimated iteratively, complicating updates and disrupting the elegant form of the boosting operator in Proposition 2. Developing theoretical guarantees for these extensions presents substantial challenges and remains an open problem.

The principles behind our methods could potentially be adapted to other machine learning frameworks, such as deep learning or ensemble methods. Exploring this adaptation could be an interesting avenue for future research. Furthermore, while Theorem 1 demonstrates that the $L_2$Boost-CUT and $L_2$Boost-IMP algorithms consistently outperform unboosted weak learners in terms of MSE, this result is established under the assumption of weak base learners (as stated in Condition (C4)). Quantifying the extent of improvement provided by boosting over unboosted learners and investigating how this improvement depends on the form of weak learners, particularly in the context of interval-censored data, would be valuable directions for future research.

## F  DETAILS OF EXPERIMENTS AND DATA IN SECTION 5

### F.1  DATA SPLITTING AND EVALUATION METRICS

The dataset is divided into $\mathcal{O}^{\text{TR}} \triangleq \{\{Y_i, X_i, \phi(X_i), u_{i,j}\} : i = 1, \ldots, n_1; j = 1, \ldots, m\}$ and $\mathcal{O}^{\text{TE}} \triangleq \{\{Y_i, X_i, \phi(X_i), u_{i,j}\} : i = n_1 + 1, \ldots, n_1 + n_2; j = 1, \ldots, m\}$ in a $4 : 1$ ratio, where $n_1 = 400$ and $n_2 = 100$. Take $\mathcal{O}^{\text{TR}}_{\text{IC}} \triangleq \{\{Y_i, X_i, u_{i,j}\} : i = 1, \ldots, n_1; j = 1, \ldots, m\}$ as training data and $\mathcal{O}^{\text{TE}}_{\text{IC}} \triangleq \{\{X_i, \phi(X_i)\} : i = n_1 + 1, \ldots, n_1 + n_2\}$ as test data. The training data $\mathcal{O}^{\text{TR}}_{\text{IC}}$ are used to obtain an estimated $\hat{f}_c$ in (2), denoted $\hat{f}^*_{n_1}$, using the proposed methods introduced in Section 2, while the test data $\mathcal{O}^{\text{TE}}_{\text{IC}}$ are employed to evaluate the performance of $\hat{f}^*_{n_1}$. For classification tasks, $\hat{f}^*_{n_1} \in [-1, 1]$, derived from the $L_2$WCBoost-based algorithm.

For regression tasks, the first metric represents the sample-based maximum absolute error (SMaxAE), defined as the infinity norm of the difference between exponential of the estimate and exponential of the true function with respect to the sample:

$$\left\| \hat{f}^*_{n_1} - \phi \right\|_\infty = \max_{X_i : i = n_1 + 1, \ldots, n_1 + n_2} \left| \exp\left\{ \hat{f}^*_{n_1}(X_i) \right\} - \exp\left\{ \phi(X_i) \right\} \right|,$$

and the second metric reports the sample-based mean squared error (SMSqE), defined as:

$$\left\| \hat{f}^*_{n_1} - \phi \right\|_2 = n_2^{-1} \sum_{i=n_1+1}^{n_1+n_2} \left[ \exp\left\{ \hat{f}^*_{n_1}(X_i) \right\} - \exp\left\{ \phi(X_i) \right\} \right]^2.$$

The smaller these metrics, the better the performance of the estimator $\hat{f}^*_{n_1}$. In addition, we consider the sample-based Kendall's $\tau$ (SKDT), defined as

$$\left\| \hat{f}^*_{n_1} - \phi \right\|_\tau = \frac{n^{\text{C}} - n^{\text{D}}}{n_2(n_2 - 1)/2},$$

where $n^{\text{C}}$ and $n^{\text{D}}$ denote the numbers of concordant and discordant pairs, respectively. For $i, i' = n_1 + 1, \ldots, n_1 + n_2$, a pair is called concordant if $\hat{f}^*_{n_1}(X_i) > \hat{f}^*_{n_1}(X_{i'})$ and $\phi(X_i) > \phi(X_{i'})$, and

discordant if $\hat{f}_{n_1}^*(X_i) \le \hat{f}_{n_1}^*(X_{i'})$ and $\phi(X_i) > \phi(X_{i'})$. This metric evaluates the concordance between $\hat{f}_{n_1}^*$ and its target function $\phi$ from a different perspective. The bigger this metric, the better the performance of the estimator $\hat{f}_{n_1}^*$.

For classification tasks, we write $\hat{f}_{n_1}^*$ as $\hat{f}_{n_1,s}^*(X_i)$ explicitly to show the dependence of the estimates on time $s$. We predict the true survival status at time $s$, with $s = 1, 2, 3$, or $4$, based on using whether $\hat{f}_{n_1,s}^*(X_i)$ is greater than 0 for $i = n_1 + 1, \ldots, n_1 + n_2$. Specifically, for $i = n_1 + 1, \ldots, n_1 + n_2$, if $\hat{f}_{n_1,s}^*(X_i) > 0$, we predict the survival status at time $s$ as 1; otherwise, we predict it as $-1$. We evaluate classification performance by using the test data $\mathcal{O}_{\mathrm{IC}}^{\mathrm{TE}}$ to calculate the *sensitivity*, defined as the proportion of correctly identified positive cases among the true positive cases, indicated by $\{i : i = n_1 + 1, \ldots, n_1 + n_2; \exp\{\phi(X_i)\} > s\}$, and the *specificity*, defined as the proportion of correctly identified negative cases among the true negative cases, indicated by $\{i : i = n_1 + 1, \ldots, n_1 + n_2; \exp\{\phi(X_i)\} \le s\}$. Sensitivity and specificity assess classification results from different perspectives. The larger these metrics, the better the performance of the estimator $\hat{f}_{n_1,s}^*$.

## F.2 LEARNING METHODS IN EXPERIMENTS

Regardless of the value of $n$, we set $w = 5$ for Algorithm 1 as a stopping criterion, and take cubic smoothing splines as base learners with $r = v = 2$ in Section 3.3. Suggested by Theorem 1, we take weak base learners. Bühlmann & Yu (2003) showed that the shrinkage strategy (Friedman, 2001) can make base learners weaker by multiplying a small constant $u$ to the smoother matrix $\Psi$. In other words, for a small constant $u$, the linear smoother learner with smoother matrix $\Psi_u = u \times \Psi$ is weaker than the linear smoother learner with smoother matrix $\Psi$. Thus, as in Bühlmann & Yu (2003), for (B.1), we set $df = 20$, and replace $\Psi$ in (14) with $\Psi_u$ and $u = 0.01$. The shrinkage strategy is equivalent to replacing Line 5 of Algorithm 1 with

$$f^{(t+1)} = f^{(t)} + u\hat{h}^{(t+1)}.$$

For ICRF, we specify the splitting rule as GWRS, described in Appendix C, adopt an exploitative survival prediction approach, use a Gaussian kernel with bandwidth $h = cn_{\min}^{-1/5}$, and take $K = 5$ and $D = 300$. Here, $c$ is the inter-quartile range of the NPMLE, and $n_{\min}$ is the minimum size of terminal nodes, set to 6.

## F.3 COMPUTING TIME COMPARISON

To access computational complexity, we record the computing time for one experiment by applying the five methods to synthetic data generated from the lognormal AFT model with $\sigma = 0.25$ in Section 5. Computing times (in second) are reported in Table F.1 for three sample sizes, where we separately display computing time for implementing ICRF from that for implementing both unbiased transformation and boosting (UT + B). The implementation of the proposed methods requires a lot longer time than that for the O, R, and N methods, as expected.

| Method
Size | O | R | N | CUT | | IMP | |
|---|---|---|---|---|---|---|---|
| | | | | ICRF | UT + B | ICRF | UT + B |
| 500 | 1.037 | 0.969 | 1.053 | 493.462 | 3.262 | 494.319 | 2.611 |
| 1000 | 2.200 | 2.190 | 2.148 | 2633.012 | 11.058 | 2668.756 | 7.935 |
| 1500 | 4.671 | 4.756 | 4.564 | 8709.231 | 18.440 | 8725.549 | 14.509 |

Table F.1: *Computing times (in second) using a cluster with 1 node, where UT and B represent the procedure corresponding to unbiased transformation and boosting, respectively.*

## F.4 SIGNAL TANDMOBIEL® DATA AND BANGKOK HIV DATA

The first dataset, called the Signal Tandmobiel® data, arose from a longitudinal prospective oral health study, conducted in the Flanders region of Belgium from 1996 to 2001. This study initially

sampled 4,430 first-year primary school children who underwent annual dental examinations performed by trained dentists. Further details can be found in Vanobbergen et al. (2000). Our analysis focuses on a subset of the data with 3737 subjects, whose features were fully observed, specifically examining the emergence times of the permanent upper left first premolars (tooth 24 in European dental notation). Following Komárek & Lesaffre (2009), we set the origin time at age 5, as permanent teeth do not emerge before this age. The response variable $Y_i$ represents the emergence time of tooth 24 since age 5 for child $i$. Because the dental examinations take place annually, the observed $Y_i$ is inherently interval censored by design. Among the participants, 1611 children are right-censored and others are truly interval-censored. The features are defined as follows: $X_{i1} = 0$ if the child is a girl and 1 otherwise; $X_{i2} = 0$ if the primary predecessor was sound, and 1 if it was decayed, missing due to caries, or filled; and $X_{i3}$ represents the scaled age at which the child started brushing teeth.

The second dataset, called Bankok HIV data, includes the information on the incidence of Human Immunodeficiency Virus (HIV) infection. To identify associated risk factors to guide prevention efforts, the Bangkok Metropolitan Administration conducted a cohort study (Vanichseni et al., 2001) in Bangkok from 1995 to 1998. The study enrolled 1124 participants who were HIV negative at the time of enrollment. These participants were repeatedly tested for HIV at approximately four-month intervals over the study period. The response variable $Y_i$ represents the time when participant $i$ was first tested positive for HIV. Among those participants, 991 were right-censored, meaning they were never tested positive during the study period, while the remaining were interval-censored, meaning the exact time of seroconversion is only known to occur between two testing intervals. The features are defined as follows: $X_{i1} = 0$ if the participant is a female and 1 otherwise; $X_{i2} = 0$ if the participant has a history of injecting drug use and 1 otherwise; and $X_{i3}$ represents the scaled age at enrollment.

## G  ADDITIONAL EXPERIMENTS

To comprehensively evaluate the performance of the proposed methods, here we conduct additional experiments to examine how their effectiveness may be influenced by various factors, including sample sizes, data generation models, noise levels, and different implementation ways of ICRF. The details are presented as follows.

### G.1  ALTERNATIVE SAMPLE SIZE AND DATA GENERATION MODEL

To assess how the sample size may affect the performance of our methods, we conduct additional experiments in the same way as in Section 5 but replace $n = 500$ by $n = 1000$. Results of predicting survival times are reported in Figure G.1, demonstrating the same patterns observed for Figure 1.

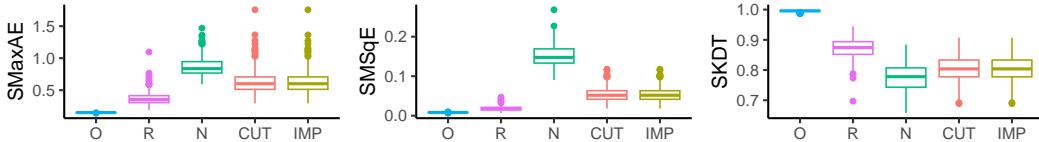

Figure G.1: *Experiment results of SMaxAE (left), SMSqE (middle), and SKDT (right) for predicting survival times with $n = 1000$, for the lognormal AFT model with $\sigma = 0.25$. O, R, N, CUT, and IMP represent the oracle, reference, naive, CUT, and IMP methods, respectively, as described in Section 5.*

In contrast to the experiment setup in Section 5, we take $p = 5$, $\tau = 12$, $m = 5$, $\phi(X_i) = \beta_0|X_{1,i} - 0.5| + \beta_1 X_{3,i}^3 + \beta_2 \sin(\pi X_{5,i})$, with $\beta_0 = 1$, $\beta_1 = 0.8$, and $\beta_2 = 0.8$, where $X_{2,i}$ and $X_{4,i}$ are inactive input variables for model (15), but they are still involved in the boosting procedure. Figure G.2 summarizes the values for the metrics, SMaxAE, SMSqE, and SKDT, across 300 experiments for predicting survival times. The N method results in the largest SMSqE yet the smallest SKDT, though the SMaxAE for the N and proposed methods are similar. Figure G.3 reports the values for two classification metrics, sensitivity and specificity, across 300 experiments for predicting survival status. The N method produces the worst results at $s = 2$ and $s = 3$, with

the lowest specificity at $s = 4$. In contrast, the proposed methods only show reduced sensitivity at $s = 4$.

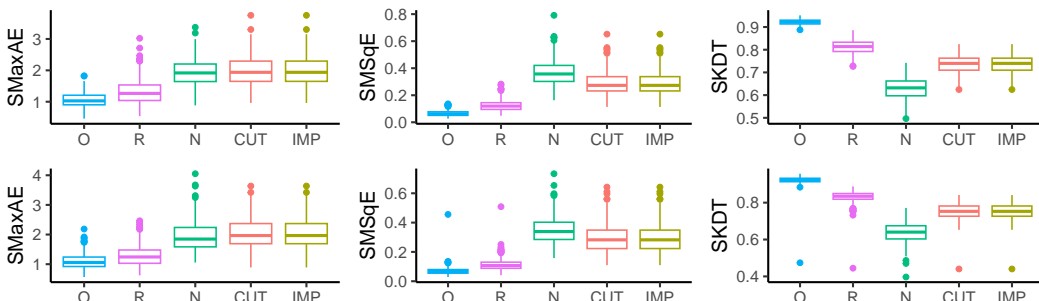

Figure G.2: *Experiment results of SMaxAE (left), SMSqE (middle), and SKDT (right) for predicting survival times with different survival models. The top and bottom rows correspond to the lognormal AFT and loglogistic AFT models, respectively. O, R, N, CUT, and IMP represent the oracle, reference, naive, CUT, and IMP methods, respectively, as described in Section 5.*

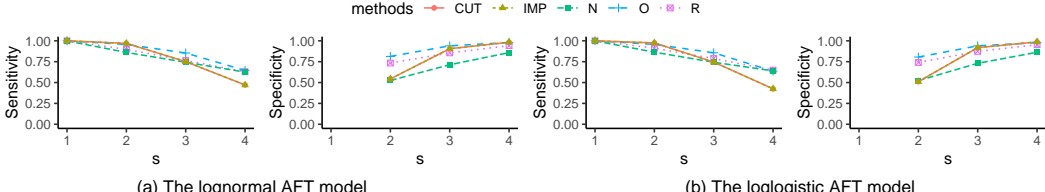

Figure G.3: *Experiment results of predicting survival status with different survival models. O, R, N, CUT, and IMP represent the oracle, reference, naive, CUT, and IMP methods, respectively, as described in Section 5. Specificity plots for $s = 1$ are omitted because no negative cases exist.*

## G.2 NOISE LEVEL COMPARISON AND COX MODEL

To access the sensitivity of our methods to the noise level of data, in addition to $\sigma = 0.25$ considered in Section 5 for model (15) with $\epsilon \sim N(0, \sigma^2)$, we further consider $\sigma = 0.5, 1$, or $1.5$. Increasing $\sigma$ makes survival times more variable, thus spanning over a wider interval. Consequently, $\tau$ is set as 15, 80, or 100 to generate interval-censored survival times. The results are reported in Figure G.4 in the same manner as for Figure 1 in Section 5. The O, R, N, CUT and IMP methods reveal the same patterns as those observed in Figure 1. The N method performs the worst, the O method performs the best, and our proposed CUT and IMP methods outperform the N method. When the noise level $\sigma$ is more substantial, the differences between our methods and the N method are considerably enlarged, and the performance of our methods becomes very close to, or nearly the same as, that of the O and R methods.

We further consider two additional methods here. The first method, denoted as YAO, employs an existing ensemble approach for interval-censored data: the conditional inference survival forest method proposed by (Yao et al., 2021), where predicted survival times are provided by the R package *ICcforest*. The results from the YAO method are in good agreement with those produced from our proposed CUT and IMP methods. However, the SKDT values from the YAO method appear slightly more variable than those from our methods.

In the second method, denoted as COX, we manipulate synthetic data to create right-censored data $\{\{\tilde{Y}_i, \Delta_i, X_i\} : i = 1, \ldots, n\}$, with pseudo-survival time $\tilde{Y}_i$ defined as in Section 5 and an artificially introduced right-censoring indicator $\Delta_i$. Here, we consider the best-case scenario where no subject is censored, with $\Delta_i$ set to 1 for all $i = 1, \ldots, n$. We then fit the data with the Cox model, where predicted survival times are taken as the medians of the estimated survivor functions by extracting the "median" column of the survfit.coxph object in the R package *survival*. While the results from the COX method are not directly comparable to the other six methods, which are primarily nonparametric-based, it is interesting that the COX method can sometimes outperform the

R method, especially when $\sigma$ is small with value 0.25, as shown by the SMaxAE and SKDT values. However, when $\sigma$ is large with value 0.5, 1 or 1.5, the COX method does not outperform the O and R methods or our proposed CUT and IMP methods, as shown by the SMaxAE and SMSqE values. Nevertheless, its SKDT values remain better than other methods, except for the O method; this may be attributed to the absence of censoring in the COX method. Suggested by the SMSqE values, the COX method can even perform worse than the N method when $\sigma$ is not small.

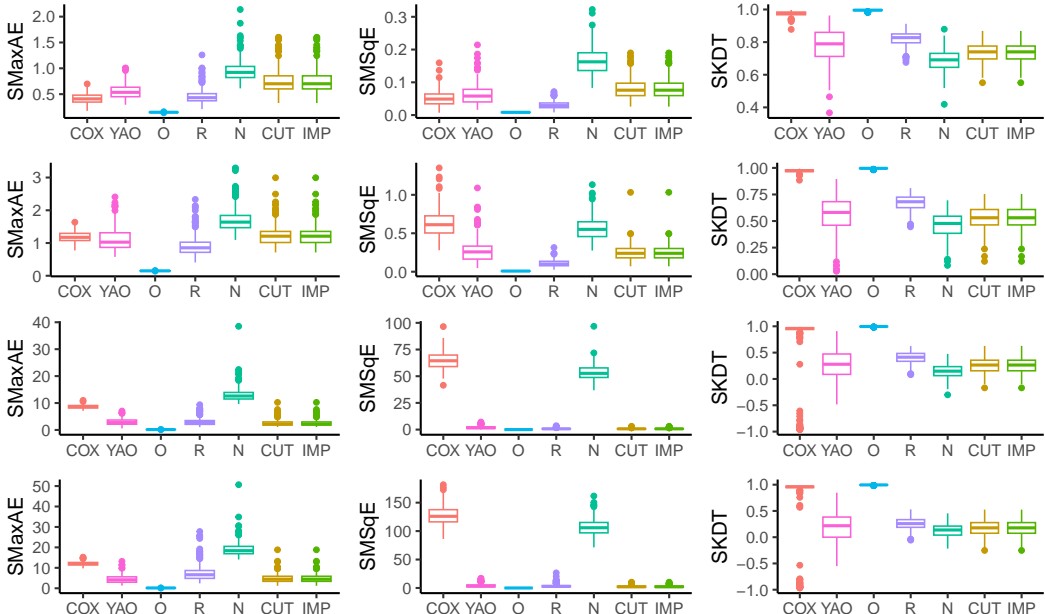

Figure G.4: *Experiment results of SMaxAE (left), SMSqE (middle), and SKDT (right), for predicting survival times with varying noise levels. From the top to bottom, the four rows correspond to the lognormal AFT model with $\sigma = 0.25$, 0.5, 1, and 1.5, respectively. COX is the procedure of fitting the Cox model to pseudo-survival times; YAO represent the method of Yao et al. (2021); O, R, N, CUT, and IMP represent the oracle, reference, naive, CUT, and IMP methods, respectively, as described in Section 5.*

### G.3 SURVIVOR FUNCTION ESTIMATOR COMPARISON

As detailed in Appendix C, the implementation of our methods employs ICRF to provide consistent estimation of the survivor function, and we take $K = 5$ and $D = 300$ to run experiments in Section 5 (as well as those additional experiments in Appendix G). To see how different choices of $K$ and $D$ may affect the performance of the proposed methods, here we implement the CUT method to synthetic data generated in Section 5 using ICRF with different values of $K$ and $D$, where we set $K = 1$ and $D = 1$; $K = 1$ and $D = 100$; $K = 1$ and $D = 300$; and $K = 3$ and $D = 300$; and we denote the resulting CUT methods as CUT1, CUT2, CUT3, and CUT4, respectively. In addition, we implement ICRF using quasi-honest survival prediction method, as discussed in Appendix C, and the comprehensive greedy algorithm (Breiman, 2001), and we denote them as CUT5 and CUT6, respectively. We report the results in Figure G.5 in the same manner as for Figure 1. The results demonstrate that the CUT method with $K = 5$ and $D = 300$ (the one with heading CUT in Figure G.5) tends to perform the best, although all other methods produce fairly close results.

## H CONVERGENCE ANALYSIS OF EXPERIMENTS

This appendix assesses the convergence of the proposed methods. For $f \in \mathcal{F}$, let $\hat{R}(f)$ denote the approximation of the empirical risk function. In Figures H.1 - H.3, we plot the values of $\hat{R}\left(f^{(t)}\right)$ and $\hat{R}\left(f_s^{(t)}\right)$ against the number of iterations $t$ for the experiments in Section 5. The results clearly

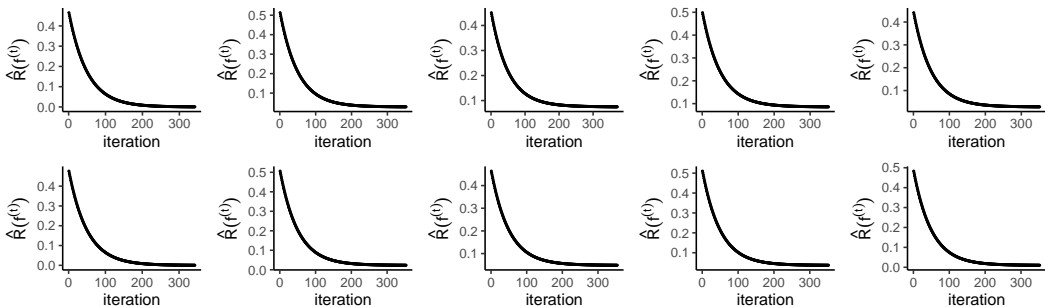

Figure G.5: *Experiment results of SMaxAE (left), SMSqE (middle), and SKDT (right) for predicting survival times with varying ICRF estimators.*

show that $\hat{R}\left(f^{(t)}\right)$ and $\hat{R}\left(f_s^{(t)}\right)$ approach zero as $t$ increases, confirming the convergence of the proposed algorithms.

Figure H.1: *Predicting survival times: Plots of $\hat{R}\left(f_s^{(t)}\right)$ versus the number of iterations. The top to bottom rows correspond to the lognormal AFT and loglogistic AFT models in Section 5, respectively. From left to right, the columns represent the O, R, N, CUT, and IMP methods, respectively. Here, O, R, N, CUT, and IMP represent the oracle, reference, naive, CUT, and IMP methods, respectively, as described in Section 5.*

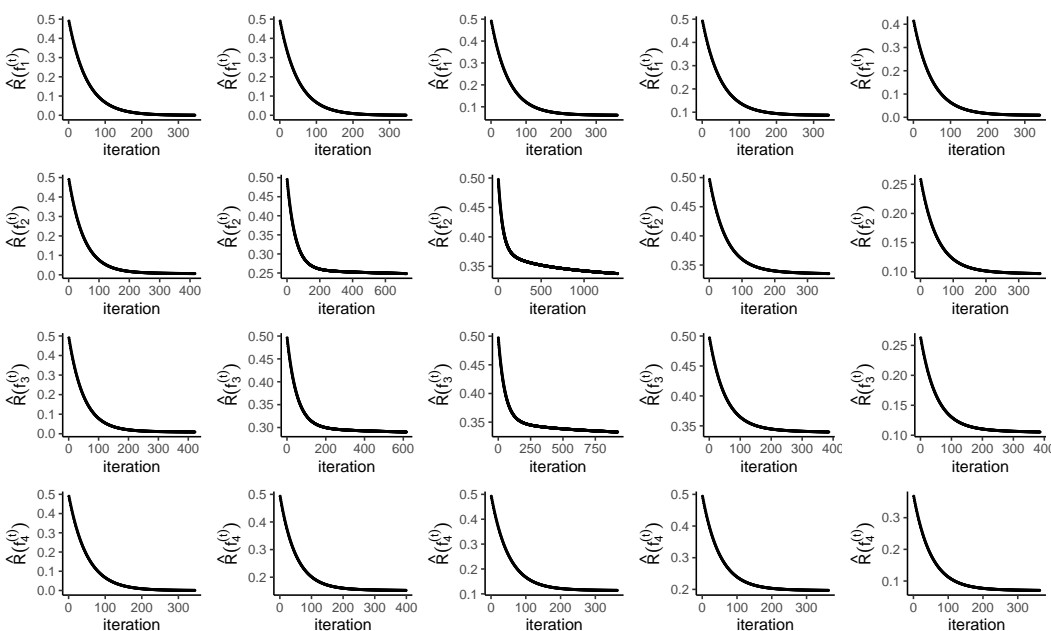

Figure H.2: *Predicting survival status – the lognormal AFT model in Section 5: Plots of $\hat{R}\left(f_s^{(t)}\right)$ versus the number of iterations. From top to bottom, each row corresponds to $s = 1, 2, 3$, and 4, respectively. From left to right, the columns correspond to the O, R, N, CUT, and IMP methods, respectively. Here, O, R, N, CUT, and IMP represent the oracle, reference, naive, CUT, and IMP methods, respectively, as described in Section 5.*

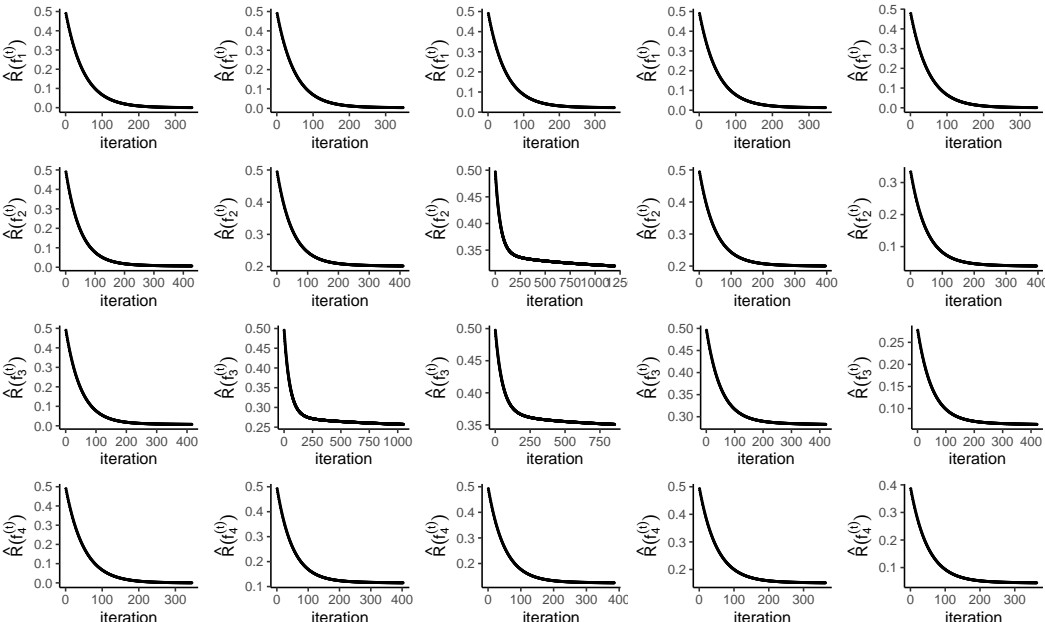

Figure H.3: *Predicting survival status – the loglogistic AFT model in Section 5: Plots of $\hat{R}\left(f_s^{(t)}\right)$ versus the number of iterations. From top to bottom, each row corresponds to $s = 1, 2, 3$, and 4, respectively. From left to right, the columns correspond to the O, R, N, CUT, and IMP methods, respectively. Here, O, R, N, CUT, and IMP represent the oracle, reference, naive, CUT, and IMP methods, respectively, as described in Section 5.*

