# OpenReview forum: "Boosting Methods for Interval-censored Data with Regression and Classification"
_ICLR.cc/2025/Conference — ICLR 2025 Poster_

### Official Review · Reviewer_Npps · 2024-10-31

**Soundness:** 3
**Presentation:** 3
**Contribution:** 3
**Rating:** 6
**Confidence:** 3

**Summary:**

The manuscript extends the boosting framework to regression and classification tasks with interval-censored data. The framework is particularly applicable in survival analysis, due to the prevalence of interval-censored data. The approach combines the L2Boost method with the censoring unbiased transformation (CUT) approach to form L2Boost-CUT, which is implemented via functional gradient descent. The manuscript also introduces an impute method based on L2Boost. L2Boost-CUT is applied to both synthetic and real data.

**Strengths:**

The paper is generally very well written and easy to follow. The authors excellently introduce the problem and previous methods. The manuscript also clearly motivates the problem. Specific strengths are:

1. The manuscript contains detailed and extensive theoretical results, providing strong theoretical guarantees.
2. The theoretical results are described very well. Each theorem and proposition is described in terms of its broader importance and the purpose for it being used. This is very important in theoretical papers, to allow the reader to follow without getting bogged down by extensive theory.
3. The manuscript tackles an interesting problem and the contribution is important. The method is applicable to many important settings.

**Weaknesses:**

My main concerns are regarding the empirical results. I'm not convinced by both the scope and outcome of the results. Specifically:

1. Both the synthetic and real data studies are very limited in scope. For an approach that is so widely useful, it is a shame that it has not been applied to more settings to showcase its usefulness. It would be insightful to find and present scenarios showing the extreme ends of performance (very good and very bad performance)  of the approach and discuss why this is.
2. The results from the studies are not overly convincing. The method performs the same as imputation and standard boosting (which already exists) and does not provide a large benefit over the naive approach in the synthetic study. I have asked a clarifying question around this in the Questions, so this weakness can be addressed.
3. I think there should be more discussion on the assumptions and limitations of the method. This would help to understand under which scenarios we might expect the method to not do well.
4. There is no discussion on computational cost or complexity. This could either be a theoretical complexity analysis or even just simple timing tests. Either would provide useful insights for practitioners who wish to decide on whether to apply the method to their problem.
5. There is no reproducibility. It would be helpful to have reproducible code, even for just a basic example. This would also partly help alleviate weakness 4, as I could run the code myself and observe the execution time.

I would like to see more evidence that the approach is beneficial in practice. I think the paper could do with some rearrangement: some of the theoretical results moved to the appendix to make space for further empirical results.

**Questions:**

1. How easily could the framework be extended to other boosting approaches (such as XG)?
2. Relating to the weakness point above: what's the benefit of L2Boost-CUT if it provides the same results as imputation + standard boosting?

---

> ### Author Response · Authors · 2024-11-25
> **Responses to Reviewer Npps (1/2)**
>
> **Strengths**:
>
> **The paper is generally very well written and easy to follow. The authors excellently introduce the problem and previous methods. The manuscript also clearly motivates the problem. Specific strengths are:**
>
> 1. **The manuscript contains detailed and extensive theoretical results, providing strong theoretical guarantees.**
>
> 2. **The theoretical results are described very well. Each theorem and proposition is described in terms of its broader importance and the purpose for it being used. This is very important in theoretical papers, to allow the reader to follow without getting bogged down by extensive theory.**
>
> 3. **The manuscript tackles an interesting problem and the contribution is important. The method is applicable to many important settings.**
>
> **Our Response**: Thank you for your positive feedback. We are delighted that you found the manuscript addresses an interesting problem and the contribution is important. We also want to thank you for dedicating your time and expertise to review our paper and provide constructive comments and suggestions. We appreciate your careful review, and we will address all comments thoroughly in our revision.
>
> **Weaknesses**:
>
> **My main concerns are regarding the empirical results. I'm not convinced by both the scope and outcome of the results. Specifically:**
>
> 1. **Both the synthetic and real data studies are very limited in scope. For an approach that is so widely useful, it is a shame that it has not been applied to more settings to showcase its usefulness. It would be insightful to find and present scenarios showing the extreme ends of performance (very good and very bad performance) of the approach and discuss why this is.**
>
> **Our Response**: Thank you for your comments and suggestions. In this revised version, we have carefully addressed your concerns by conducting additional experiments and data analyses to explore diverse settings.
>
> First, we have now included an additional real-world dataset to further validate the practical utility of our methods. The analysis results are now presented  in Figure 3 and discussed at the end of Section 5.
>
> Second,  we have now expanded synthetic experiments for various scenarios in Appendix G, including varying levels of noise, sample sizes, and data generation models, to provide a more comprehensive understanding of the performance of our proposed methods.
>
> Furthermore, as ICRF comprises a component in our proposed methods (clarified at the end of Section 3.3), we have now conducted further experiments to evaluate how different implementations of ICRF influence performance of our proposed methods. Results and discussions are now included in Appendix G.3.
>
> 2.  **The results from the studies are not overly convincing. The method performs the same as imputation and standard boosting (which already exists) and does not provide a large benefit over the naive approach in the synthetic study. I have asked a clarifying question around this in the Questions, so this weakness can be addressed.**
>
> **Our Response**: Thank you for your comments, which we have now carefully addressed. In our initial version, we did not include comparison details due to space limitations. In this revision, we have added more information to compare the two algorithms, explaining their differences and similarities, in Section 3.2 for greater clarity.
>
> Furthermore, in this revised manuscript, we have conducted additional experiments; details can be found in Appendix G. The results demonstrate the practical utility of our proposed methods.
>
> 3. **I think there should be more discussion on the assumptions and limitations of the method. This would help to understand under which scenarios we might expect the method to not do well.**
>
> **Our Response**: To address your comments, we have now included the discussion in Section 6.
>
> 4.  **There is no discussion on computational cost or complexity. This could either be a theoretical complexity analysis or even just simple timing tests. Either would provide useful insights for practitioners who wish to decide on whether to apply the method to their problem.**
>
> **Our Response**: Thank you for highlighting the importance of discussing computational complexity. In response, in this revision we have now included the discussion in Appendix E.2. In addition, in this revision we have re-ran experiments and added computational timing comparisons for better insights into finite sample implementation. These results are now reported in Table F.1 of Appendix F.3.

---

> > ### Author Response · Authors · 2024-11-25
> > **Responses to Reviewer Npps (2/2)**
> >
> > 5.  **There is no reproducibility. It would be helpful to have reproducible code, even for just a basic example. This would also partly help alleviate weakness 4, as I could run the code myself and observe the execution time.**
> >
> > **Our Response**: Thank you for highlighting the importance of reproducibility and suggesting the inclusion of reproducible code. We plan to make the code publicly available on GitHub upon the paper's acceptance. To address this, we have added a note after Proposition 1 in Section 3.2 to inform readers of our intention.
> >
> > 6. **I would like to see more evidence that the approach is beneficial in practice. I think the paper could do with some rearrangement: some of the theoretical results moved to the appendix to make space for further empirical results.**
> >
> > **Our Response**: Thank you for the constructive suggestion. In this revised version, we have carefully organized the material to ensure it is both informative and comprehensive.
> >
> > **Questions**:
> >
> > **How easily could the framework be extended to other boosting approaches (such as XG)? Relating to the weakness point above: what's the benefit of L2Boost-CUT if it provides the same results as imputation + standard boosting?**
> >
> > **Our Response**: Thank you for your thoughtful comments and questions. In this revision, we have carefully  addressed them. First, we respond to your comment on the extensibility to other boosting approaches. While our focus in this paper is on $L_2$Boost with the CUT-based loss function, the general principle of incorporating interval-censored data into boosting frameworks is not limited to $L_2$Boost, and the proposed framework can be adapted to other boosting methods. Depending on the form of loss functions, some extension are straightforward with theoretical guarantees readily established by modifying our theoretical derivations. For other loss functions, while the implementation procedures can be readily formed by modifying Algorithm 1, developing sound theoretical guarantees is not straightforward, which warrants in-depth explorations. Specifically, in this revision we have included the details in Appendix E.3.
> >
> > Regarding the comparison between $L_2$Boost-CUT and $L_2$Boost-Impute, we have now make this more clear with the revisions after Proposition 1 in Section 3.2.

---

> > > ### Comment · Reviewer_Npps · 2024-11-25
> > >
> > > I appreciate the effort and time taken to respond to my review. I have considered your responses carefully and will leave my score as it is.

---

> > > > ### Author Response · Authors · 2024-11-26
> > > > **thanks**
> > > >
> > > > Dear Reviewer Npps:
> > > >
> > > > Thank you very much for your prompt response to our rebuttal. We deeply appreciate your time and insightful comments, which have greatly helped improve the presentation of our work.

---

### Official Review · Reviewer_VgwR · 2024-10-31

**Soundness:** 4
**Presentation:** 2
**Contribution:** 2
**Rating:** 5
**Confidence:** 2

**Summary:**

The boosting method is widely used in machine learning field, but it is not obvious to extend it to interval-censored data, that is, the outcome value is not specifically given but its interval is given. To apply the boosting to such datasets, the paper applies the method called the "censoring unbiased transformation" (CUT) to the loss function. As described in Proposition 1, the application of CUT does not change the expected loss between the interval-censored data and the true (uncensored) data, if we know the conditional survivor function $S$ (in reality we need to estimate it from the dataset). With this conversion of the problem, they showed that we can apply L2boost, an existing method of boosting for regression datasets (it can be extended to binary classifications).

**Strengths:**

-   The paper found that, by CUT, existing L2boost algorithm can be applied to interval-censored datasets.
    -   There is a constraint that we need to estimate the joint distribution of $X$ and $Y$ without knowing $Y$ itself but knowing only intervals. They showed that estimating the distribution by ICRF experimentally worked well.
-   Theoretical convergence rates and lower bounds of MSE (for regressions) and misclassifications (for classifications) are presented.

**Weaknesses:**

-   The procedure itself looks somewhat simple; first we apply CUT and then L2boost. If there is a difficulty or interesting results of these combinations, please emphasize.
    -   Perhaps, the combination of boosting and interval-censored data is the novelty? If so, please emphasize the discussion on the novelty (e.g., limitations in existing methods).

**Questions:**

Key questions

-   It uses ICRF to estimate the conditional survivor function, but perhaps can we just impute interval-censored outcomes by ICRF (instead of using (Bian et al., 2024a))?
-   Section 1.1: The paper presented several tree-based interval-censored regression methods, but is it difficult to extend these methods to boosting? If so, why?
-   Section 1.2: It states that L2Boost-Impute is a proposed method, but the procedure appears only in Section 5, and the imputation method is just a reference. What is the novelty?
-   Section 5, Figure 1: Why the methods "O" (oracle) and "R" (reference) are compared in these plots? It is true that these results uses unknown information in reality and they can produce better results, but it looks that these results are not discussed in the paper.

Minor questions and suggestions

-   Some overlaps of letters found, so please consider replacing either of them with another letter.
    -   $L$: overlapped between the "loss function" and the "left of the interval"
    -   $S$: overlapped between the "conditional survivor function" and the "smoother matrix"
-   Section 3.1, line 182: Should $L = l, R = r, l < Y \leq r$ be $L = l, R = r, L < Y \leq R$, as far as reading the succeeding equation? (Does it mean that the distribution of $Y$ depends on the observation of the random variables $L$ and $R$ but independent of the distributions of $L$ and $R$?)
-   Section 3.1, line 187: It looks that the randomness of $M$ is not used anywhere else. How does the randomness work?
-   Section 3.2, line 238: It states that we need to estimate $S(y|X_i)$, but the procedure has not been described at this point. Please consider referring Section 3.3 as the section that describes how to estimate it.
-   Section 5: The "sample-based maximum absolute error" is abbreviated as "S-MAE", but please consider another abbreviation, since "MAE" also stands for **mean** absolute error. (It is confusing since "S-MSE" represents the "sample-based **mean** squared error".)
-   Section 5, line 471: adpots --> adopts
-   Section 5, line 510: boxlplots --> boxplots
-   Section 5, Figure 1(b): lines for "CUT" are not visible since they look overlapped with other lines, so please change line style or transparency so that the lines appear.

---

> ### Author Response · Authors · 2024-11-25
> **Responses to Reviewer VgwR (1/3)**
>
> **Strengths**:
> - **The paper found that, by CUT, existing L2boost algorithm can be applied to interval-censored datasets.**
> - **There is a constraint that we need to estimate the joint distribution of $X$ and $Y$ without knowing $Y$ itself but knowing only intervals. They showed that estimating the distribution by ICRF experimentally worked well.**
> - **Theoretical convergence rates and lower bounds of MSE (for regressions) and misclassifications (for classifications) are presented.**
>
> **Our Response**: Thank you for dedicating your time and expertise to review our paper and provide constructive comments and suggestions. We appreciate your careful review, and we will address all comments thoroughly in our revision.
>
> **Weaknesses**:
> - **The procedure itself looks somewhat simple; first we apply CUT and then L2boost. If there is a difficulty or interesting results of these combinations, please emphasize.**
> - **Perhaps, the combination of boosting and interval-censored data is the novelty? If so, please emphasize the discussion on the novelty (e.g., limitations in existing methods).**
>
> **Our Response**: Thank you for your comments and suggestions, which have helped us revise the paper and more clearly articulate the contributions of our work.
> The key innovation of our work lies in presenting a framework that extends boosting methods to handle interval-censored data — a crucial yet underexplored problem in machine learning. Standard boosting algorithms are not directly applicable in this context due to the challenges posed by interval censoring. Our approach addresses this gap by leveraging the censoring-unbiased transformation (CUT) to construct unbiased loss functions tailored for interval-censored data.
>
> Motivated by your comment and feedback from other reviewers, we have revised the paper to include extensions of our framework to other loss functions that can be used to handle interval-censored data. We have also highlighted the uniqueness of the $L_2$ loss function employed in our current theoretical development and emphasized that establishing similar theoretical results for these extensions is not straightforward.
>
> In response to your suggestion, we have included a discussion on possible extensions in Appendix E.3 as well as the limitations of our methods in Section 6.

---

> > ### Author Response · Authors · 2024-11-25
> > **Responses to Reviewer VgwR (2/3)**
> >
> > **Questions**:
> >
> > **Key questions**
> > - **It uses ICRF to estimate the conditional survivor function, but perhaps can we just impute interval-censored outcomes by ICRF (instead of using (Bian et al., 2024a))?**
> >
> > **Our Response**: Thank you for your question. ICRF was designed to estimate the conditional survival function (Cho et al., 2022), but it does not directly predict survival time or status, which is the primary focus of our paper.
> >  While it is possible to use ICRF to impute interval-censored outcomes, we chose to incorporate ICRF as part of our $L_2$Boost-CUT and $L_2$Boost-IMP methods. These two-step approaches enable us to leverage the strengths of both ICRF for imputing the interval-censored data and $L_2$Boost for subsequent predictions. For greater clarity, in this revision we have included the discussions in Sections 3.2 and 3.3.
> > - **Section 1.1: The paper presented several tree-based interval-censored regression methods, but is it difficult to extend these methods to boosting? If so, why?**
> >
> > **Our Response**: Thank you for the comments. Yao et al. (2021) introduced a survival forest method utilizing the conditional inference framework, while Cho et al. (2022) proposed a recursive forests method. These methods are specifically designed for estimating survival functions for interval-censored data and cannot directly predict survival time or status. In contrast, our proposed boosting method builds upon survival function estimation as a foundational step, where Yao et al. (2021) and Cho et al. (2022)'s methods can be incorporated as a basic component of our framework.
> >
> > Yang et al. (2024) adopted a similar approach by constructing an observed loss function and developing tree algorithms for interval-censored data. However, their focus differs from ours as they emphasize tree-based methods, while we concentrate on boosting combined with smoothing splines. Moreover, our work offers comprehensive theoretical results for boosting methods, whereas Yang et al. (2024) provided only the implementation procedure without theoretical justification.
> >
> > Extending tree-based methods to boosting may present a challenge due to the iterative refinement of model parameters in boosting, where the learning rate and loss function need to be carefully optimized. In our proposed $L_2$Boost-CUT and $L_2$Boost-IMP methods, we leverage the unique properties of the $L_2$ loss, enabling a simpler boosting algorithm with the optimal learning rate inherently incorporated into the optimization process for $\hat{h}$. While tree-based methods, such as those proposed by Yao et al. (2021) and Cho et al. (2022), can be adapted procedurally to a boosting framework with a loss function other than the $L_2$ loss, establishing theoretical guarantees remains challenging and uncertain.
> >
> > To address your comments, in this revision, we have included additional discussions on this aspect in Appendix E.3. As a side note, in this revision we have run additional experiments using the method by Yao et al. (2021), as suggested by another reviewer. The results, reported in Figure G.4 of Appendix G.2, shows that the results from the Yao et al. (2021) method are in good agreement with those produced from our proposed methods. However, the SKDT (Kendall's $\tau$) values from this method appears slightly more variable than those from our methods.
> > -  **Section 1.2: It states that L2Boost-Impute is a proposed method, but the procedure appears only in Section 5, and the imputation method is just a reference. What is the novelty?**
> >
> > **Our Response**: Thank you for raising this careful comment, which we have now carefully addressed. In our initial version, we did not include comparison details due to space limitations. In this revision, we have added more information to compare the two algorithms, explaining their differences and similarities in Section 3.2 for greater clarity.
> >
> > - **Section 5, Figure 1: Why the methods O (oracle) and R (reference) are compared in these plots? It is true that these results uses unknown information in reality and they can produce better results, but it looks that these results are not discussed in the paper.**
> >
> > **Our Response**:
> > As you correctly pointed out, the O (oracle) and R (reference) methods rely on information unavailable in real-world scenarios and thus cannot be applied in actual data analysis. The inclusion of these methods in Figure 1 serves to provide benchmarks for our proposed methods, illustrating the upper bounds of performance for a realistic method. Including these benchmarks allows us to assess how our methods perform relative to those designed for ideal conditions with full data availability.

---

> > > ### Author Response · Authors · 2024-11-25
> > > **Responses to Reviewer VgwR (3/3)**
> > >
> > > **Minor questions and suggestions:**
> > > - **Some overlaps of letters found, so please consider replacing either of them with another letter: $L$ overlapped between the loss function and the left of the interval; $S$ overlapped between the conditional survivor function and the smoother matrix.**
> > >
> > > **Our Response**: Thank you for this careful comment. In preparing a revised manuscript, we will use different notation to differentiate them from greater clarity.
> > > - **Section 3.1, line 182: Should $L=l, R=r, l < Y \le r$ be $L=l, R=r, L < Y \le R$ as far as reading the succeeding equation? (Does it mean that the distribution of $Y$ depends on the observation of the random variables $L$ and $R$ but independent of the distributions of $L$ and $R$?)**
> > >
> > > **Our Response**:
> > > Thank you very much for insightfully pointing out this error. Our intended meaning was as you perceived, but it was not precisely presented. We have now corrected it.
> > > - **Section 3.1, line 187: It looks that the randomness of $M$ is not used anywhere else. How does the randomness work?**
> > >
> > > **Our Response**: Thank you for this insightful question. While the randomness of $M$ may not be directly used in specific calculations, it serves a crucial role in our framework by illustrating that our methods are designed to accommodate settings where the number of observations is not fixed but can vary with uncertainty. This flexibility is important in practical applications, where real-world data may  not have a rigid, predefined number of observations.
> > >
> > > In real applications with a single dataset, $M$ is realized  as the number of observations, but treating $M$ as a random variable highlights our methods’ robustness across scenarios with differing numbers of observations.
> > > To clarify, we have updated the text to better explain $M$.
> > > - **Section 3.2, line 238: It states that we need to estimate $S(y|X_i)$, but the procedure has not been described at this point. Please consider referring Section 3.3 as the section that describes how to estimate it.**
> > >
> > > **Our Response**: Thank you for this careful suggestion. In this revision, we have addressed this point by referring to Section 3.3.
> > > - **Section 5: The sample-based maximum absolute error is abbreviated as S-MAE, but please consider another abbreviation, since MAE also stands for {\bf mean} absolute error. (It is confusing since S-MSE represents the sample-based mean squared error.)**
> > >
> > > **Our Response**: Thank you for your comments on the abbreviations S-MAE and S-MSE. To avoid potential confusion with MAE (mean absolute error) and MSE (mean squared error), we have now  revised the abbreviations. Specifically, S-MAE has been replaced with SMaxAE,  and S-MSE has been replaced with SMSqE.
> > > - **Section 5, line 471: adpots -> adopts**
> > >
> > > **Our Response**: Thank you for pointing out this typo. We have fixed it for the revision.
> > > - **Section 5, line 510: boxlplots -> boxplots**
> > >
> > > **Our Response**: Thank you for pointing out this typo. We have fixed it for the revision.
> > > - **Section 5, Figure 1(b): lines for CUT are not visible since they look overlapped with other lines, so please change line style or transparency so that the lines appear.**
> > >
> > > **Our Response**: Thank you for this suggestion. In response, we have updated Figure 1(b) and the related figures by modifying the line and point styles to ensure the CUT lines are clearly visible and no longer overlap with other lines.

---

> ### Comment · Reviewer_VgwR · 2024-11-25
>
> Thank you for rigorous responses and updates.
> However, it is still unclear for me about the **benefit of employing boosting** in the proposed problem setup, although the method itself becomes clear for me.
> It would be the best if the performance outperformed existing methods, but it looks not true as far as reading the experimental results (although certain improvements are seen). However, it is good if there are good insights through the proposed method, for example, "this is an interesting result in the field of boosting" or "this methodology can be extended to other learning methods than boosting". How do the authors think?

---

> > ### Author Response · Authors · 2024-11-27
> >
> > Thank you for your thoughtful feedback on our rebuttal. We would like to further address your comments and clarify the broader contributions  of our work.
> >
> > #    **Performance Compared to Existing Methods:**
> >
> > Regarding your observation,  "It would be the best if the performance outperformed existing methods, but it looks not true as far as reading the experimental results (although certain improvements are seen)", we agree that this is an important point. These observations reflect the unique challenges posed by interval-censored data.
> >
> > To the best of our knowledge, our methods are the first to handle **predictions for various features** related to survival processes with **interval-censored data**. As such, there are no existing methods available for **direct, fair comparisons**. The only meaningful baselines are the N, O, and R methods described in Section 5, as their differences can be clearly delineated and compared. While we have included additional experiments incorporating the YAO and COX methods for baseline comparisons in the revised manuscript, these methods do not share comparable characteristics for fair evaluation.
> >
> > Theoretically, our Proposition 1 demonstrates that the censoring unbiased transformation (CUT)-based $L_{CUT}$​ loss function has the same risk as the original $L_2$​ loss. However, determining the transformed outcomes in  (7) adds complexity due to the need for survival function estimation. Moreover, the absence of direct measurements of survival times introduces substantial variability, as evidenced by the noticeable differences between our methods and the R method, which assumes access to true responses. Nonetheless, our methods show significant improvement over the naive method, which incorporates boosting but fails to properly account for the effects of interval-censoring.
> >
> > # **Benefit of Boosting for Interval-Censored Data:**
> >
> > In survival analysis, particularly with interval-censored data, survival times are often only partially observed, creating significant analytical challenges. Boosting offers advantages for addressing these challenges:
> >
> > (1)  **Information Aggregation:**
> >
> > Boosting combines weak learners into a strong ensemble, effectively aggregating the limited and incomplete information that can be extracted from interval-censored datasets. This characteristic is particularly valuable for survival data, where individual methods often struggle to extract sufficient signal from severely incomplete data.
> >
> > (2) **Enhanced Information Extraction:**
> >
> > Our framework leverages the boosting technique in conjunction with censoring-specific strategies such as the censoring unbiased transformation (CUT) and imputation (IMP). These methods integrate into the boosting process to improve the ability to utilize the partial information inherent in interval-censored data.
> >
> > # **Insights and Broader Contributions:**
> >
> > We appreciate your suggestion to emphasize the broader implications and insights of our work. The principles behind our methods could be adaptable to other machine learning frameworks, such as deep learning or ensemble methods, broadening their applicability beyond boosting. This can be an interesting project for future exploration. Furthermore, while Theorem 1 demonstrates that the $L_2$Boost-CUT and $L_2$Boost-IMP algorithms consistently outperform unboosted weak learners in terms of MSE, this result is established under the assumption of weak base learners (as per Condition (C4)). Inspired by your comments, it would be valuable to quantify the extent of improvement offered by boosting over unboosted learners and to investigate how this improvement depends on the form of weak learners, particularly in the context of interval-censored data.
> > In preparing the final version of our manuscript, we plan to add these aspects  for possible future research.
> >
> > # **Conclusion:**
> >
> > In summary, our work demonstrates the practical value of employing boosting in survival analysis with interval-censored data by effectively addressing the unique challenges posed by incomplete information. The methodology we propose not only improves information extraction in this domain but also offers theoretical insights and paves the way for future extensions to other learning paradigms.
> >
> >
> > Thank you once again for your constructive feedback and thoughtful suggestions, which we deeply appreciate. They have been invaluable in improving the presentation and expanding the scope of our manuscript. Your precious time and valuable insights are deeply appreciated.

---

> > > ### Comment · Reviewer_VgwR · 2024-11-27
> > >
> > > I was convinced with the responses, especially with the section "Benefit of Boosting for Interval-Censored Data". The effectiveness of ensemble methods for interval-censored datasets, and the development of a boosting method for such datasets are valuable.

---

> ### Author Response · Authors · 2024-11-27
>
> Dear Reviewer VgwR:
>
> Many thanks for your prompt response. We value your comments, suggestions, and feedback, which greatly help us  sharpen the presentation of our work. With kind regards.

---

### Official Review · Reviewer_uzby · 2024-11-01

**Soundness:** 3
**Presentation:** 4
**Contribution:** 3
**Rating:** 6
**Confidence:** 4

**Summary:**

The manuscript introduces a framework that extends boosting techniques to effectively handle interval-censored data. The authors offer a thorough theoretical analysis, examining mean squared error (MSE), variance, and bias, building on foundational results from Bühlmann & Yu (2003). The proposed methods are further substantiated through experiments conducted on both synthetic and real-world datasets, demonstrating the framework’s applicability and effectiveness.

**Strengths:**

- The manuscript is clearly written and accessible, making the methodology easy to follow.
- The authors conduct a rigorous theoretical analysis of their proposed methods, assessing mean squared error (MSE), variance, and bias to establish a solid foundation for their approach.

**Weaknesses:**

- In Proposition 2, using a distinct notation for the smoother matrix would help prevent confusion with the survival function $S$.
- Adding experiments for the Cox proportional hazards model would strengthen the manuscript's applicability and support its conclusions.
- Additional ensemble methods exist for interval-censored data, such as:
    - Yao, W., Frydman, H., and Simonoff, J.S., 2021. An ensemble method for interval-censored time-to-event data. Biostatistics, 22(1), pp.198-213.

  Including comparisons with such methods would enhance the evaluation of the proposed approach.
- In both synthetic and real data experiments, the proposed CUT method and the imputation approach (Bian et al. 2024a) yield comparable results, with the imputation method occasionally performing better (e.g., in real data). Could the authors further elaborate on the unique advantages of their proposed method?
- In Appendix A, the authors outline several conditions primarily related to the smoother matrix $S$. Providing relevant examples of $S$ that meet these conditions would improve clarity.
- All theoretical results assume consistent estimation of the survival function. How does the convergence rate of the survival function estimator impact the final estimator?

**Questions:**

- In page 3 line 108, should the survival probability be P(Y>s)?
- In page 3 Equation (5), what is the update compared to Equation (3)?
- In traditional boosting methods, a learning rate is typically included. Is there a similar parameter in Algorithm 1?
- In Condition (C1), what does $Q_i^2$ mean? Does it denote $Q_i'Q_i$ as $Q_i$ is an eigenvector?

---

> ### Author Response · Authors · 2024-11-25
> **Responses to Reviewer uzby**
>
> **Strengths**:
> - **The manuscript is clearly written and accessible, making the methodology easy to follow.**
> - **The authors conduct a rigorous theoretical analysis of their proposed methods, assessing mean squared error (MSE), variance, and bias to establish a solid foundation for their approach.**
>
> **Our Response**: Thank you for your encouraging feedback. We are pleased  that you found the manuscript clear and accessible, and we appreciate your recognition of the rigorous theoretical analysis. We also want to thank you for dedicating your time and expertise to review our paper and provide constructive comments and suggestions. We appreciate your careful review, and we have addressed all comments thoroughly in our revision.
>
> **Weaknesses**:
> - **In Proposition 2, using a distinct notation for the smoother matrix would help prevent confusion with the survival function $S$**
>
> **Our Response**: Thank you for this suggestion. In the revision, we have now incorporated a different symbol, $\Psi$, for improved clarity.
>
> - **Adding experiments for the Cox proportional hazards model would strengthen the manuscript's applicability and support its conclusions.**
>
> **Our Response**: In response to your suggestion, we have included results derived from the Cox model in both data analyses and experiments. These results are now presented in Figure 3 (Section 5) and Figure G.4 (Appendix G.2), along with corresponding comments in the respective sections.
>
> - **Additional ensemble methods exist for interval-censored data, such as: Yao, W., Frydman, H., and Simonoff, J.S., 2021. An ensemble method for interval-censored time-to-event data. Biostatistics, 22(1), pp.198-213. Including comparisons with such methods would enhance the evaluation of the proposed approach.**
>
> **Our Response**: Thank you for this suggestion, which has now been incorporated in this revision. Specifically, additional experiment results,  reported in Figure G.4 in Appendix G.2, shows that the results from the Yao et al. (2021) method are in good agreement with those produced from our proposed methods. However, the SKDT (Kendall's $\tau$) values from this method appears slightly more variable than those from our methods.
>
> - **In both synthetic and real data experiments, the proposed CUT method and the imputation approach (Bian et al. 2024a) yield comparable results, with the imputation method occasionally performing better (e.g., in real data). Could the authors further elaborate on the unique advantages of their proposed method?**
>
> **Our Response**: Thank you for raising this interesting point, which we have now carefully addressed. In our initial version, we did not include comparison details due to space limitations. In this revision, we have added more information to compare the two algorithms, explaining their differences and similarities in Section 3.2 for greater clarity.
>
> - **In Appendix A, the authors outline several conditions primarily related to the smoother matrix $S$.  Providing relevant examples of $S$ that meet these conditions would improve clarity.**
>
> **Our Response**: Thank you for the suggestion. In this revision, we have now inserted the following material at the end of Appendix A.
>
> - **All theoretical results assume consistent estimation of the survival function. How does the convergence rate of the survival function estimator impact the final estimator?**
>
> **Our Response**: Thank you for raising this important concern. We agree that the convergence rate of the survival function estimator  can affect the speed at which the estimator approaches the true value.  A slower convergence rate may increase variability in the final estimator, potentially resulting in less reliable predictions, particularly in finite sample settings. To make this point clear, in this revision we have now included discussions in Section 3.3.
>
> **Questions**:
> - **In page 3 line 108, should the survival probability be $P(Y>s)$?**
>
> **Our Response**: Thank you. We have now fixed this.
> - **In page 3 Equation (5), what is the update compared to Equation (3)?**
>
> **Our Response**: For greater clarity, we have revised the presentation of Equation (5) by combining it with the follow-up description originally in Equation (6).
> - **In traditional boosting methods, a learning rate is typically included. Is there a similar parameter in Algorithm 1?**
>
> **Our Response**: Thank you for this insightful question. To make this point clear, in this revision we have now included discussions in Appendix E.1.
> - **In Condition (C1), what does $Q_i^2$ mean? Does it denote $Q_i^\prime Q_i$ as $Q_i$** as is an eigenvector?
>
> **Our Response**: Thank you very much for pointing out this error. You are correct; it was meant to be $Q_i^\top Q_i$. We have now corrected it.

---

> > ### Comment · Reviewer_uzby · 2024-11-25
> >
> > In the revised manuscript, the authors state: "We analyze synthetic data using the proposed L2Boost-CUT (CUT) and L2Boost-IMP (IMP) methods, as opposed to three other methods." However, after a careful comparison of the original and revised manuscripts, it seems that the so-called L2Boost-IMP (IMP) method corresponds to the imputation approach proposed by Bian et al. (2024a) in the original draft. Could the authors clarify why this discrepancy exists?
> >
> > Furthermore, in Section 3.2, the authors mention:
> > - "These two algorithms are derived from distinct perspectives in addressing interval-censored data, and they may be expected to yield different results."
> > - "As a result, L2Boost-CUT and L2Boost-IMP differ mainly in the stopping criterion, suggesting that they may often yield similar results."
> >
> > If L2Boost-IMP is indeed the imputation method proposed in Bian et al. (2024a), this raises questions about the novelty of the proposed methods. Specifically, if the differences between the two methods lie only in the stopping criterion, and the results are often similar or marginally different, it becomes unclear what new insights or advantages the proposed method offers compared to the existing approach. As such, I see no compelling reason to revise my current score.

---

> ### Author Response · Authors · 2024-11-26
>
> Thank you for providing more detailed comments. Here, we further clarify and address the concerns you raised.
>
> **1. The differences between Bian et al. (2024a) and our L2Boost-IMP (IMP)**
>
> We would like to clarify that our L2Boost-IMP method is fundamentally different from the approach proposed by Bian et al. (2024a). The reference to Bian et al. (2024a) in the Introduction is included solely to provide context about boosting methods in diverse applications.
>
>  Both the L2Boost-IMP method and Bian et al.'s (2024a) approach share a common use of *imputation* to create complete datasets for applying algorithms designed for full data. However, key differences between the methods pose distinct challenges in establishing their respective theoretical guarantees, as highlighted below.
>
> **(a) different contexts:**
>
> The method proposed by Bian et al. (2024a) was developed for missing responses, where a simple missing data indicator $\Delta_i$ ($\Delta_i= 1$ if response $Y_i$ is observed, and 0 otherwise) suffices to reflect the missingness status for each subject $i$. In contrast, our L2Boost-IMP (IMP) method is tailored for interval-censored responses, which require a more complex representation. Instead of a scalar missing data indicator $\Delta_i$,  a sequence of censoring indicators $\Delta_{i,j}$  is needed for each subject $i$, corresponding to different intervals indexed by $j$.
>
> **(b) different constructions of pseudo-outcomes:**
>
> Bian et al. (2024a) used the Buckley-James  formulation (Buckley and James 1979) by defining a pseudo-outcome as
> $$ Y_i^* \triangleq \Delta_i Y_i+(1-\Delta_i) E(Y_i|X_i, \Delta_i=0). $$
> However,  our approach constructs the pseudo-outcome differently, as shown in (7), which incorporates the survivor function.
>
> **(c) different implementations:**
>
> In determining the pseudo-outcome $Y_i^*$, Bian et al. (2024a) approximated the conditional expectation $E(Y_i|X_i, \Delta_i=0)$ using  the Monte-Carlo method. They generated a sequence of variates
>    $y_i^{(k)}$ from the conditional distribution $f(y_i|X_i, \Delta_i=0)$ for $k =1, \ldots, K$, which $K$ is a user-specified, and computed
> the average $(1/K)\sum_{k=1}^K y_i^{(k)}$. On the contrary,
> determining the pseudo-outcome in our approach, as outlined in (7), requires estimating the survivor function, making the implementation of our $L_2$Boost-IMP (IMP) method considerably more complex than that of Bian et al. (2024a).
>
> **(d) different properties of imputed outcome/loss function:**
>
> The pseudo-outcomes in Bian et al. (2024a)  satisfy the properties
> $$ E(Y^*_i|X_i)=E(Y_i|X_i), \ \ \mbox{and thus}, \ \ E(Y^*_i)=E(Y_i). $$
>
> However, pseudo-outcomes in our approach lack this  property due to the complexity introduced by the interval-censoring structure of the data.
>
> **2. differences between  our proposed two methods: L2Boost-CUT (CUT) and  L2Boost-IMP (IMP)**
>
> The CUT-based loss function $L_{\rm CUT}({\cal O}_i, f(X_i))$ in (9) modifies the $L_2$ loss, denoted $L(u,v)$,  to ensure Proposition 1 holds. However, this adjusted loss function   is  **quadratic** in the difference between its first and second arguments, meaning  it is not identical to the $L_2$ loss  with its first argument imputed by the transformed response
>  $\tilde Y_1({\cal O}_i)$ in (7).
>
> Specifically  $$     L_{\rm CUT}({\cal O}_i, f(X_i)) \ne L(\tilde Y_1({\cal O}_i),f(X_i)).   $$
>
> This distinction underpins our earlier statement that "These two algorithms are derived from distinct perspectives in addressing interval-censored data, and they may be expected to yield different results."
>  The $L_2$Boost-CUT method focuses on **adjusting the loss function** so its expectation recovers that of the original  $L_2$ loss  $L$, whereas the  $L_2$Boost-IMP  method **preserves** the functional form of the original loss $L$ but **replaces** its first argument with  transformed response $\tilde Y_1({\cal O}_i)$ in (7).
>
>  Despite this difference, since the $L_2$ loss has a derivative linear in its first argument, the increment terms in both $L_2$Boost-CUT and $L_2$Boost-IMP are closely related. However, if the loss function were $L_q$  with $q\ge 3$, the development of
> $L_q$Boost-CUT and $L_q$Boost-IMP  methods would differ more substantially. However, as noted in Appendix E.3, establishing theoretical guarantees for such cases is  challenging.
>
> For a general loss function, particularly when it is nonlinear, constructing an adjusted loss function like $L_{\rm CUT}$ ​ in (9) to ensure Proposition 1 holds is challenging, making it difficult to implement. However, using the original loss function with imputed values determined by the transformed response in (7) is a  straightforward implementation, though establishing theoretical results for both cases remains difficult.
>
> Thank you for your thoughtful question on this aspect. We will incorporate these clarifications in the next revision.

---

### Official Review · Reviewer_mmSc · 2024-11-01

**Soundness:** 3
**Presentation:** 3
**Contribution:** 3
**Rating:** 5
**Confidence:** 3

**Summary:**

In this work the authors propose a new algorithm based on boosting for interval-censored data. They provide an extensive theoretical analysis and an empirical comparison of both simulated and real data, for which their proposed methods have a performance similar to that of the state of the art.

**Strengths:**

The article is clearly written and provides a thorough theoretical analysis of the proposed algorithm, which is really important to ground further applicative work using this method in practice.

**Weaknesses:**

There are several limitations of this work.
- my main concern is that the paper does not clarify the empirical improvement compared to the use of ICRF. It is slightly different indeed to use boosted trees compared to random forests (bagging), but the article fails to show a clear gain in performance, both on simulated and real data
- there should be more baselines, like Cox models, even if they are designed for right-censored data rather than interval data. There are very few baselines included in the benchmark presented here. It's unclear whether the I method is similar to ICRF or a novel method. If not, ICRF should be included as a baseline to demonstrate the contributions of this work clearly.
- I don't think there is any mention that the code to use L2Boost-CUT or L2Boost-Impute will be made available, is it?

**Questions:**

see limitations
- are there any additional performance metrics that could be used? like the c-index (or a variation of it)?

---

> ### Author Response · Authors · 2024-11-25
> **Responses to Reviewer mmSc**
>
> **Strengths**:
>
> **The article is clearly written and provides a thorough theoretical analysis of the proposed algorithm, which is really important to ground further applicative work using this method in practice.**
>
> **Our Response**: Thank you for your positive feedback and for recognizing the importance and thoroughness of our work. We also want to thank you for dedicating your time and expertise to review our paper and provide constructive comments and suggestions. We appreciate your careful review, and we will address all comments thoroughly in our revision.
>
> **Weaknesses**:
>
> **There are several limitations of this work.**
> - **my main concern is that the paper does not clarify the empirical improvement compared to the use of ICRF. It is slightly different indeed to use boosted trees compared to random forests (bagging), but the article fails to show a clear gain in performance, both on simulated and real data.**
>
> **Our Response**: Thank you for raising this concern. We would like to clarify the distinction of our work in relation to ICRF. ICRF (Cho et al., 2022) is a tree-based, nonparametric method specifically designed for estimating survival functions for interval-censored data, which differs from our objective. We focus on developing boosting methods for regression and classification tasks with interval-censored data. Specifically, we aim to construct a predictive model $f(\cdot)$ that well predicts a transformed target variable $g(Y)$, where $g(\cdot)$ can take various forms to address different tasks. These tasks include predicting the survival time, the survival status at specific time points, or any functional outcome of survival time of interest.
>
> Our method builds upon survival function estimation as a basic step, for which ICRF is utilized as a component of our framework. Therefore, our work is not designed to improve ICRF but rather to leverage it within a broader algorithm to achieve new predictive objectives. Consequently, the goal of our paper is distinct from ICRF’s, and direct numerical comparisons between the two would not be meaningful. For greater clarity, in this revision we have included some discussions at the end of Section 3.3 to address your comment, as well as a related comment from another reviewer.
>
> - **there should be more baselines, like Cox models, even if they are designed for right-censored data rather than interval data. There are very few baselines included in the benchmark presented here. It's unclear whether the I method is similar to ICRF or a novel method. If not, ICRF should be included as a baseline to demonstrate the contributions of this work clearly.**
>
>
> **Our Response**: Thank you for your feedback. Regarding your comment on ICRF, as explained in the response to the previous bullet point,  the I method (now referred to as the L2Boost-IMP method, or IMP for short) is not similar to ICRF. Instead, ICRF constitutes a component of the I method and is used specifically for estimating the survivor function.
>
> In response to your suggestion regarding the use of the Cox model, we have included additional results derived from it in the revised manuscript. Specifically, we now present results for both the data analysis and experimental settings involving the Cox model. They now appear in Figure 3 in Section 5, as well as Figure G.4 in Appendix G.2.
>
> - **I don’t think there is any mention that the code to use L2Boost-CUT or L2Boost-Impute will be made available, is it?**
>
> **Our Response**: Thank you for your comment. We plan to make the code publicly available after acceptance and will post it on GitHub at that time. In this revision, we have added a note after Proposition 1 in Section 3.2 to inform readers of this intention.
>
> **Questions:**
> - **are there any additional performance metrics that could be used? like the c-index (or a variation of it)?**
>
> **Our Response**: Thank you for your thoughtful suggestion regarding additional performance metrics. To address this, we have incorporated Kendall’s $\tau$ to evaluate the concordance between the estimator and its target. Kendall’s $\tau$ quantifies the ordinal association between two variables by comparing concordant and discordant pairs. This metric captures both the consistency of the predicted ranking with the true ranking and the extent of any discrepancies between them. Alongside the metrics considered in the initial version, we now present experimental results using Kendall’s $\tau$ in Figure 1, as well as Figures G.1. G.2, G.4, and G. 5 in Appendix G. Additionally, we have included the discussions in Appendix F.1, Section 5, and Appendix G.1.

---

> > ### Comment · Reviewer_mmSc · 2024-11-25
> >
> > The authors provided interesting answers and explained some parts of the paper that were still slightly unclear to me. I am still not entirely convinced by the contribution, as it seems that most of the performance is due to the use of another algorithm to impute the "missing" data, and then the application of rather standard machine learning tools. However, my initial grade does not reflect the quality of the paper, so I increase my grade to 5.
> > I regret that the code was not submitted in an anonymous way so that it could be reviewed with the paper. One of the main interest of the article is to contribute to define good practice when using censored data, which can not be done with a bad code or no code.

---

> > > ### Author Response · Authors · 2024-11-26
> > >
> > > We appreciate your thoughtful feedback and for taking the time to review our rebuttal. We are grateful for your increased assessment of our work.  Below, we would like to further address your remaining concerns in more detail.
> > >
> > > 1.  **Regarding the Contribution and Performance Dependence on Imputation:**
> > >
> > > We would like to address the perception of our contributions: "as it seems that most of the performance is due to the use of another algorithm to impute the `missing' data, and then the application of rather standard machine learning tools."
> > >
> > > While we agree that the proposed $L_2$Boost-IMP employs the  **imputation** as a first step, the novelty of our approach lies in its tailored design for the **unique challenges posed by interval-censored data**. Unlike standard imputation techniques, our approach ensures that the imputed values are consistent with the underlying data distribution. Furthermore, the integration of these imputed values with machine learning tools  provides a robust framework for  handling interval-censored data. This combination goes beyond a simple `impute-and-apply' approach.
> > >
> > > Handling imputation for interval-censored responses is significantly more complex than standard imputation for missing responses. To draw an analogy, this extension mirrors the difference between calculus and arithmetic: while calculus builds upon the foundational elements of arithmetic, it represents a substantial conceptual and methodological leap. Similarly, our approach is not a straightforward application of standard imputation and machine learning tools but rather a novel framework designed to address the intricacies of interval-censored data.
> > >
> > > To further clarify, we highlight the differences between our $L_2$Boost-IMP and the usual procedure for imputation of missing responses:
> > >
> > >    (a) **Complexity in Facilitating Missingness:**
> > >
> > > In the context with missing responses, a single missing data indicator, say  $\Delta_i$ ($\Delta_i= 1$ if response $Y_i$ is observed, and 0 otherwise), suffices to reflect the missingness status for each subject $i$. However, to facilitate interval-censored responses,  our $L_2$Boost-IMP  method  requires a more complex representation. Instead of a scalar missing data indicator
> > > $\Delta_i$,  a sequence of censoring indicators $\Delta_{i,j}$  is needed for each subject $i$, corresponding to different intervals indexed by $j$.
> > >
> > > (b) **Complexity in Constructing of Pseudo-Outcomes:**
> > >
> > > In the context with missing responses, an imputation model based on the conditional distribution $f(y_i|X_i, \Delta_i=0)$ is often
> > > used to determine an imputed value for a missing response. However, our $L_2$Boost-IMP  method requires a more complex approach. Determining the pseudo-outcome, as outlined in (7), involves estimating the survivor function,  which significantly increases the complexity of implementing the $L_2$Boost-IMP  method compared to standard imputation procedures.

---

> > > > ### Author Response · Authors · 2024-11-26
> > > >
> > > > 2. **Our Contributions Go Beyond Introducing the $L_2$Boost-IMP  Method:**
> > > >
> > > > Our work  presents a comprehensive framework that significantly advances the applicability of boosting methods to interval-censored data -- a critical yet underexplored challenge in machine learning. Key contributions include:
> > > >
> > > >   (a)  **Two Complementary Methods:**
> > > >
> > > > In addition to introducing the $L_2$Boost-IMP  method, we also propose the $L_2$Boost-CUT method approaching the problem from a different perspective. This method focuses on **adjusting the loss function** so its expectation recovers that of the original  $L_2$ loss  $L$, whereas the  $L_2$Boost-IMP  method **preserves**  the functional form of the original loss $L$ but **replaces** its first argument with  transformed response $\tilde Y_1({\cal O}_i)$ in (7).
> > > >
> > > >   (i)  **Difference Between the Two Methods:**
> > > >
> > > >  The CUT-based loss function $L_{\rm CUT}({\cal O}_i, f(X_i))$ in (9) modifies the $L_2$ loss, denoted $L(u,v)$,  to ensure Proposition 1 holds. Because this adjusted loss function   is  **quadratic** in the difference between its first and second arguments,  it is not identical to the $L_2$ loss  with its first argument imputed by the transformed response  $\tilde Y_1({\cal O}_i)$ in (7), as used in the $L_2$Boost-IMP method. That is,
> > > >
> > > >  $$  L_{\rm CUT}({\cal O}_i, f(X_i)) \ne L(\tilde Y_1({\cal O}_i),f(X_i)). $$
> > > >
> > > >    (ii)  **Connection Between the Two Methods:**
> > > >
> > > >  Despite this difference, we identify a unique connection between the two proposed methods. Due to the linear derivative of the $L_2$ loss with respect to  its first argument, the increment terms in both $L_2$Boost-CUT and $L_2$Boost-IMP are closely related. This connection reflects the underlying coherence between the two approaches, despite their distinct formulations.
> > > >
> > > >  (b) **Theoretical Justifications:**
> > > >
> > > >   Our work goes beyond merely providing boosting procedures to handle interval-censored data; it also offers rigorous theoretical justifications, including evaluations of mean squared error (MSE), variance, and bias.  These analyses significantly enrich the literature on boosting methods, particularly for data with complex censoring structures.
> > > >
> > > >  (c) **Possible Extensions:**
> > > >
> > > >  For general loss functions, particularly nonlinear ones, constructing an adjusted loss function like $L_{\rm CUT}$ in (9) to ensure Proposition 1 holds is challenging and, in many cases, impractical. However, extending the idea of $L_2$Boost-IMP -- using the original loss function with imputed values determined by the transformed response in (7) -- provides a straightforward implementation framework, as suggested in Appendix E.3.
> > > >
> > > > (d)  **Summary:**
> > > >
> > > > Our approaches tackle the unique challenges of interval-censored data from complementary angles:
> > > >
> > > >    (i)  The $L_2$​Boost-CUT method introduces a novel framework by **adjusting the loss** function, while the $L_2$​Boost-IMP method **leverages imputation**  with transformed responses to retain the loss's original form.
> > > >
> > > >    (ii) Both methods ensure that imputed values align with the complex **interval-censoring** structure of the data.
> > > >
> > > >    (iii) Our **theoretical contributions and practical methodologies** collectively extend the frontiers of machine learning for censored data, providing a robust framework for broader application across various domains.
> > > >
> > > > 3.  **Regarding the Submission of Code:**
> > > >
> > > > We share your view on the importance of code availability in ensuring transparency and reproducibility. Due to the constraints of the anonymous review process, we were unable to include our code at the time of submission. However, we are fully committed to making our code publicly available upon acceptance, ensuring it meets high-quality standards and provides clear implementation details.
> > > >
> > > > We hope these clarifications address your remaining concerns and highlight the novelty and practical value of our contributions.  We will further incorporate these clarifications when preparing the final version of the manuscript. Thank you once again for your thoughtful comments and constructive feedback, which have greatly improved the presentation and clarity of our work. Your time and expertise are deeply appreciated.

---

> > > > > ### Author Response · Authors · 2024-11-27
> > > > >
> > > > > Dear Reviewer mmSc:
> > > > >
> > > > > Again, many thanks for your feedback on our rebuttal. We share your view on the importance of code availability. We have now uploaded our raw code files as supplementary materials, which include four files: readme.md (provides information about the other three files), fun.R (contains helper functions), cut.R (implements the CUT algorithm), and imp.R (implements the IMP algorithm used in the experiments in Section 5). A user-friendly version of the code, with self-contained explanations and better documentation of implementation details, will be made available on GitHub upon acceptance of the paper
> > > > >
> > > > > We value your comments, suggestions, and feedback, which greatly help us sharpen the presentation of our work.
> > > > >
> > > > > With kind regards.

---

### Official Review · Reviewer_4h4i · 2024-11-03

**Soundness:** 3
**Presentation:** 4
**Contribution:** 3
**Rating:** 6
**Confidence:** 3

**Summary:**

The paper presents boosting algorithms for regressors and classifiers  for the censored data. The censoring naturally introduces a bias into the estimator as it collapses  the mass of the tails of the underlying distribution on the censoring boundary.  In particular, the main contribution of the paper is in presenting the novel boosting framework with censoring unbiased transformation which focuses on modifying the loss function. Authors also present a model for imputing the missing data in censoring context. Authors investigate the theoretical properties of the algorithms and conclude that incorporating the spline estimators into the base learner results in optimal MSE rates for the predictor.

**Strengths:**

- The paper is mathematically well written. The notation is consistent and precise. Definitions and proofs are provided in the Appendix.
- The theoretical results are novel, clearly structured and formulated. The implications of each theoretical result are well discussed and framed within the literature context if relevant.
- The proposed algorithms are experimentally tested on the synthetic dataset under various scenarios. The method is further applied to real-life dataset.

**Weaknesses:**

1) The underlying part of the model is estimating survival function. It would be interesting to include the experiment with different survival function estimators to assess how sensitive the method is to the potential biases of the estimator of  the survival times as this part of the model is not properly covered by presented theory.
2) It is not clear how sensitive the proposed method is to the noise in the underlying data.
3) Authors did not provide much insights into how well the algorithm scales to the larger datasets.

**Questions:**

1) What are the scaling capabilities of the models to the large datasets? (provided datasets are of order of magnitude 10^3? It would be good to see experiments for varying sizes of the dataset.
2) The imputation algorithm and the CUT boosting in the presented experiments (especially Figure 3, E.3) seems to produce similar results. Can you provide more insights into why that is?
3) It would be great to see performance for other real-life datasets with already known benchmarks to demonstrate that the presented method performs well against other well explored survival dataset problems.
4) For the synthetic dataset, e.g. scenario 1, how does the performance changes based on the amount of the noise in the data, so when you change the variance in the error term $epsilon_i$, e.g. instead of only 0.25 variance for normal, when you increase the noise for to 0.5, 1., 1.5?

---

> ### Author Response · Authors · 2024-11-25
> **Responses to Reviewer 4h4i**
>
> **Strengths**:
>  - **The paper is mathematically well written. The notation is consistent and precise. Definitions and proofs are provided in the Appendix.**
>  - **The theoretical results are novel, clearly structured and formulated. The implications of each theoretical result are well discussed and framed within the literature context if relevant.**
>  - **The proposed algorithms are experimentally tested on the synthetic dataset under various scenarios. The method is further applied to real-life dataset.**
>
> **Our Response**: Thank you for the positive feedback. We are glad you recognize the novelty of the theoretical results and the mathematical rigor of the paper. We also want to thank you for dedicating your time and expertise to review our paper and provide constructive comments and suggestions. We appreciate your careful review, and we have carefully addressed all comments thoroughly in our revision.
>
> **Weaknesses**:
> 1. **The underlying part of the model is estimating survival function. It would be interesting to include the experiment with different survival function estimators to assess how sensitive the method is to the potential biases of the estimator of the survival times as this part of the model is not properly covered by presented theory.**
>
> 2. **It is not clear how sensitive the proposed method is to the noise in the underlying data.**
>
> 3. **Authors did not provide much insights into how well the algorithm scales to the larger datasets.**
>
> **Our Response**: In this revision, we have carefully addressed each of your comments to improve upon the weakness of our initial submission. Specifically, in response to your suggestion about survivor function estimation, we have now conducted additional experiments and reported the results in Figure G.5, with detailed comments provided in Appendix G.3. Regarding your comments on the sensitivity of our methods to noise level and data size, we have thoroughly addressed them. Details are provided in the following bullet points, corresponding to your individual questions.
>
> **Questions**:
> 1. **What are the scaling capabilities of the models to the large datasets? (provided datasets are of order of magnitude $10^3$? It would be good to see experiments for varying sizes of the dataset.**
>
> **Our Response**: Thank you for raising this important question. To address your concern, in this revision we have conducted additional experiments using a larger dataset with $n=10^3$ to evaluate the scalability of our methods. Results for these experiments have now been summarized in Figure G.1 in Appendix G.1.
>
> 2. **The imputation algorithm and the CUT boosting in the presented experiments (especially Figure 3, E.3) seems to produce similar results. Can you provide more insights into why that is?**
>
> **Our Response**: Thank you for raising this interesting point, which we have now carefully addressed. In our initial version, we did not include comparison details due to space limitations. In this revision, we have added more information to compare the two algorithms, explaining their differences and similarities. Specifically, we have inserted the following material in Section 3.2 for greater clarity.
>
> 3. **It would be great to see performance for other real-life datasets with already known benchmarks to demonstrate that the presented method performs well against other well explored survival dataset problems.**
>
> **Our Response**: Thank you for this suggestion. We agree that benchmarking the proposed methods on multiple survival datasets would demonstrate the broader applicability of our approaches.
> In response to your suggestion, we have included another dataset -- the Bangkok HIV data -- in this revision and analyzed it using the proposed methods. The results are now presented alongside the initial analysis of the Signal Tandmobiel dataset in Figure 3 in Section 5 (Here, results of implementing the Cox model are also included in response to other reviewers' suggestion).
>
> 4. **For the synthetic dataset, e.g. scenario 1, how does the performance changes based on the amount of the noise in the data, so when you change the variance in the error term, e.g. instead of only 0.25 variance for normal, when you increase the noise for to 0.5, 1, 1.5?**
>
> **Our Response**: Thank you for the suggestion. In this revision, we have incorporated your feedback
>  by conducting additional experiments to  extend the initial analysis with $\sigma=0.25$ to include $\sigma=0.5$, $\sigma=1$,  and $\sigma=1.5$. The results are summarized in Figure G.4, together with some comments presented in Appendix G.2.

---

> > ### Comment · Reviewer_4h4i · 2024-11-27
> >
> > I would like to thank authors for their response, addressing my concerns and adding additional work. I am happy with the manuscript. Releasing code would be definitely great. I will keep my score as it is.

---

> > > ### Author Response · Authors · 2024-11-27
> > >
> > > Dear Reviewer 4h4i:
> > >
> > > Many thanks for your feedback on our rebuttal. We share your view on the importance of code availability. We have now uploaded our raw code files as supplementary materials, which include four files: readme.md (provides information about the other three files), fun.R (contains helper functions), cut.R (implements the CUT algorithm), and imp.R (implements the IMP algorithm used in the experiments in Section 5). A user-friendly version of the code, with self-contained explanations and better documentation of implementation details, will be made available on GitHub upon acceptance of the paper
> > >
> > > We value your comments, suggestions, and feedback, which greatly help us sharpen the presentation of our work.
> > >
> > > With kind regards.

---

### Author Response · Authors · 2024-11-25
**Consolidated Rebuttal to All Reviewers**

Dear reviewers,

Thank you for dedicating your valuable time and effort to reviewing our paper. We appreciate your insightful feedback and constructive comments, which have greatly helped us enhance the presentation of our work. We are grateful for your recognition of the contributions and strengths of our paper, as summarized below:
- The manuscript tackles an **interesting problem and the contribution is important**. The method is applicable to many important settings.
- **The theoretical results are novel, clearly structured and formulated**. The implications of each theoretical result are **well discussed and framed** within the literature context if relevant.
- The article provides a **thorough theoretical analysis** of the proposed algorithm, which is really **important to ground further applicative work using this method in practice**.
- The paper is **mathematically well written**. The notation is consistent and precise. The paper is  easy to follow. The authors **excellently introduce** the problem and previous methods. The manuscript also clearly motivates the problem.
- **The theoretical results are described very well**. Each theorem and proposition is described in terms of its broader importance and the purpose for it being used. **This is very important in theoretical papers**, to allow the reader to follow without getting bogged down by extensive theory.
-  The proposed algorithms are **experimentally tested** on the synthetic dataset under various scenarios. The method is further applied to real-life dataset.

Regarding the weakness of our article,  we have carefully reviewed each of your queries, concerns, and comments. In preparing the revised
version, we have thoroughly addressed each comment to improve the clarity and presentation of our paper. Here, we highlight the key changes made in this revision. Detailed explanations for the changes we have made to each specific comment are provided in the responses to each reviewer.
- Responses to the query about the **similarity of the imputation algorithm and the CUT boosting** (from **Reviewers 4h4i, uzby, VgwR, Npps**):

In our initial version, we did not include comparison details due to space limitations. In this revision, we have added detailed descriptions  to compare the two algorithms, explaining their differences and similarities in Section 3.2 for greater clarity.
- Responses to the **clarification  of the proposed methods and ICRF** (from **Reviewers mmSc, uzby, VgwR**):

Our method builds on survival function estimation, with ICRF utilized as a component of our framework. Therefore, our work is not intended to improve ICRF but to leverage it within a broader algorithm to achieve new predictive objectives. As a result, the goal of our paper differs from that of ICRF, and direct numerical comparisons between the two would not be meaningful. For clarity, we have included some discussions at the end of Section 3.3 in this revision.
- Responses to the suggestion  of **including additional baseline**, like the Cox model and the method by Yao et al. (2021) (from **Reviewers mmSc, uzby**):

We have included additional comparison results in the revised manuscript. Specifically, for data analysis, we now present results using the Cox model in Figure 3 of Section 5. Results from experiments involving the Cox model and the method by Yao et al. (2021) are now shown in Figure G.4 in Appendix G.2.
- Responses to the comments on **sensitivity** of the proposed methods (from **Reviewer  4h4i**) and the use of **additional metric** to summarize experiment results (from **Reviewer mmSc**):

To address these comments, we have now conducted additional experiments and reported the results using the additional metric, Kendall's $\tau$ (denoted SKDT in the manuscript), along with  the initial metrics. Please see its definition in Appendix F.1, and the results reported in Figure 1 of Section 5 and Figure G.4 in Appendix G.2, together with the related comments in those places.
- Responses to the comments on **extensions and limitations** of our methods (from **Reviewers VgwR, Npps**):

In response to these comments, we have included possible extensions in Appendix E.3 and revised Section 6 to discuss the limitations of our methods.

Finally, we have carefully revised the manuscript to make the presentation more concise and informative. Key messages are presented in the main text, while lengthy technical details, discussions, and additional experiments are provided in the appendices with clear headings for each topic. We have also proofread the paper to correct spelling errors. For your convenience, we have marked the major revisions in red.

Thank you for your constructive comments and suggestions, which have greatly enhanced our manuscript.

---

> ### Author Response · Authors · 2024-11-28
>
> Dear reviewers:
>
> Thank you all for your continued feedback on our rebuttal and for your helpful suggestions. In addition to  providing our further responses to your additional feedback individually,  we have revised the manuscript once again to incorporate your  comments for greater clarity. The changes made in response to your initial feedback are marked in red, while the new revisions are highlighted in blue for ease of your reference.
>
> We deeply appreciate your time and valuable expertise, which have significantly improved the presentation of our work. We believe this revised version represents a considerable improvement over the initial draft for greater clarity.
>
> Thank you once again for your support.
>
> With warm regards

---

### Meta-Review · Area_Chair_6XsQ · 2024-12-17

**Metareview:**

This paper has been borderline in the review process, with a short range of rating but most reviewers judging its content good. In addition to taking into account the rebuttal process, I read the paper.

Regarding the rebuttal process, I personally appreciated the attention to details in the authors' reply to uzby on the differences with the work of Bian et al. (2024a). As seen from the updated draft, the authors have made very substantial updates to their draft during the rebuttal process, which is very appreciated. I appreciate the additional work on noise handling, in which the authors also have added more baselines. And more metrics on separate experiments. The particular care given overall to the reply to all specific comments made by reviewers stands out.

Now, regarding the paper's content, I found it both interesting since the problem is indeed non trivial for boosting, and the treatment of the problem quite well done in a work with good balance between theory and experiments. I only regret that the paper does not dig into boosting properties as first designed in Valiant's model, but this is a detail. I trust the authors will make their code available. I appreciated the attention to details given in the theory section and the extensive experiments with many baselines used.

In the end, taking into account the author's care in their responses during rebuttal, the numerous updates in the draft, I consider this paper worthy of presentation.

**Additional Comments On Reviewer Discussion:**

The authors have done an excellent job of replying to each reviewer, tackling many different subjects / properties and being very specific on each of them, which made reading / judging / comparing very easy.

---

### Decision · Program_Chairs · 2025-01-22

Accept (Poster)